# Brain-like Flexible Visual Inference by Harnessing Feedback-Feedforward Alignment

**Tahereh Toosi** [*]
Center for Theoretical Neuroscience
Zuckerman Mind Brain Behavior Institute
Columbia University
New York, NY
tahereh.toosi@columbia.edu

**Elias B. Issa**
Department of Neuroscience
Zuckerman Mind Brain Behavior Institute
Columbia University
New York, NY
elias.issa@columbia.edu

## Abstract

In natural vision, feedback connections support versatile visual inference capabilities such as making sense of the occluded or noisy bottom-up sensory information or mediating pure top-down processes such as imagination. However, the mechanisms by which the feedback pathway learns to give rise to these capabilities flexibly are not clear. We propose that top-down effects emerge through alignment between feedforward and feedback pathways, each optimizing its own objectives. To achieve this co-optimization, we introduce Feedback-Feedforward Alignment (FFA), a learning algorithm that leverages feedback and feedforward pathways as mutual credit assignment computational graphs, enabling alignment. In our study, we demonstrate the effectiveness of FFA in co-optimizing classification and reconstruction tasks on widely used MNIST and CIFAR10 datasets. Notably, the alignment mechanism in FFA endows feedback connections with emergent visual inference functions, including denoising, resolving occlusions, hallucination, and imagination. Moreover, FFA offers bio-plausibility compared to traditional back-propagation (BP) methods in implementation. By repurposing the computational graph of credit assignment into a goal-driven feedback pathway, FFA alleviates weight transport problems encountered in BP, enhancing the bio-plausibility of the learning algorithm. Our study presents FFA as a promising proof-of-concept for the mechanisms underlying how feedback connections in the visual cortex support flexible visual functions. This work also contributes to the broader field of visual inference underlying perceptual phenomena and has implications for developing more biologically inspired learning algorithms.

## 1  Introduction

Humans possess remarkable abilities to infer the properties of objects even in the presence of occlusion or noise. They can mentally imagine objects and reconstruct their complete forms, even when only partial information is available, regardless of whether they have ever seen the complete form before. The process of visual inference on noisy or uncertain stimuli requires additional time, implying cognitive processes that go beyond a simple feedforward pass on visual input and suggest the involvement of additional mechanisms such as feedback and recurrence (Kar et al., 2019; Kietzmann et al., 2019; Gilbert and Sigman, 2007; Debes and Dragoi, 2023; Kreiman and Serre, 2020). Despite the abundant evidence on the involvement of feedback connections in various cognitive processes, understanding the precise mechanisms through which they flexibly give rise to the ability to infer or generate perceptual experiences is not clear.

---

[*]Correspondence to: Tahereh Toosi <tahere.toosi@gmail.com>

37th Conference on Neural Information Processing Systems (NeurIPS 2023).

While hierarchical feedforward models of the ventral visual cortex based on deep learning of discriminative losses have achieved remarkable success in computer vision tasks (Yamins et al., 2014; Khaligh-Razavi and Kriegeskorte, 2014; Lindsay, 2021), alternative frameworks, such as predictive processing models, offer a distinct perspective on visual processing. Predictive processing models propose that the brain generates fine-grained predictions about incoming sensory inputs and compare them with actual sensory signals to minimize prediction errors (Rao and Ballard, 1999; Friston, 2009; Clark, 2013). These models emphasize the role of feedback connections in the visual cortex, with higher-level areas sending top-down predictions to lower-level areas to guide perception (Wen et al., 2018; Choksi et al., 2021; Whittington and Bogacz, 2017; Millidge et al., 2020). Unlike deep learning models that learn from large-scale datasets, predictive processing models prioritize the role of prior knowledge and expectations in shaping perception as originally emphasized as far back as Helmholtz (Helmholtz et al., 1909). While predictive processing models emphasize the active role of top-down predictions and prior knowledge in shaping visual perception, they lag behind feedforward models in mechanistic specificity and thus direct neurophysiological evidence (Walsh et al., 2020; Clark, 2013; Keller and Mrsic-Flogel, 2018). Moreover, most large-scale implementation of predictive coding or, in general, recurrence relies on back-propagation-through-time (BPTT) to train the recurrence and feedback connections (Choksi et al., 2021; Tscshantz et al., 2023; Spoerer et al., 2017), which suffers from many issues regarding bio-plausible implementation (Lillicrap and Santoro, 2019a).

Another main constraint on the space of models is the learning algorithm to train the model. Classical error backpropagation (BP) has been a workhorse algorithm for training discriminative, feedforward deep neural networks, particularly for visual object recognition (Rumelhart et al., 1986; Krizhevsky et al., 2012). Despite its immense success in training the state-of-the-art, BP has been criticized on a number of implementational issues, some of which also call into question its bio-plausibility for the brain: weight symmetry requires weight transport to the feedback network (Grossberg, 1987), the feedback network is not used during runtime inference, and feedforward discrimination performance is not robust to noise (Goodfellow et al., 2015; Akrout, 2019). Beyond these issues, BP is an infinitesimally local estimate of the gradient, and other higher-order methods for computing the gradient could accelerate learning. As pointed out by (Bengio, 2014), the inverse of the weight matrix, rather than the transpose, may provide a valid path for credit assignment, learning a linear extrapolation of the underlying landscape (Bengio, 2014). However, attempts to match BP by learning the inverse weights instead of the transpose in a stage-wise fashion, also called target propagation (TP), have yielded limited practical success for reasons that are not entirely clear (Lee et al., 2014; Bartunov et al., 2018) – potentially related to the difficulty of learning an inverse function using noisy gradients as opposed to the relative ease of taking a transpose, a noiseless procedure (Kunin et al., 2020).

Here, we simultaneously learn feedforward and feedback functions that are mutual global inverses of each other such that each path can perform credit assignment for the other during the training pass. We term this Feedback-Feedforward Alignment (FFA) since the discriminator (encoder) contributes the gradients for the reconstructor (decoder) and vice versa. We show that rather than trading against each other as in typical, single-objective settings, co-optimizing discrimination and reconstruction objectives can lead to a mutualistic symbiotic interaction.

Next, we explore the potential of the gradient path as a model of feedback connections. We are inspired by the structural similarity of the credit assignment computational graph and the feedforward pass, which parallels the anatomically reciprocated feedforward and feedback connections in the primate visual cortex (Markov et al., 2013, 2014). Importantly we hypothesized an objective function for feedback connections motivated by the *High-resolution buffer hypothesis* by (Lee and Mumford, 2003) regarding the primary visual cortex (V1), arguing that V1 is uniquely situated to act as a high-resolution buffer to synthesize images through generative processes. We hypothesized that by co-tuning feedforward and feedback connections to optimize two different but dependent objective functions, we could explain the properties of flexible visual inference of visual detail under occlusion, denoising, dreaming, and mental imagery.

The contributions of this study are as follows:

- Based on the role of feedback connections in the brain, we propose a novel strategy to train neural networks to co-optimize for two objective functions.

- We leverage the credit assignment computational graph as feedback connections during learning and inference.

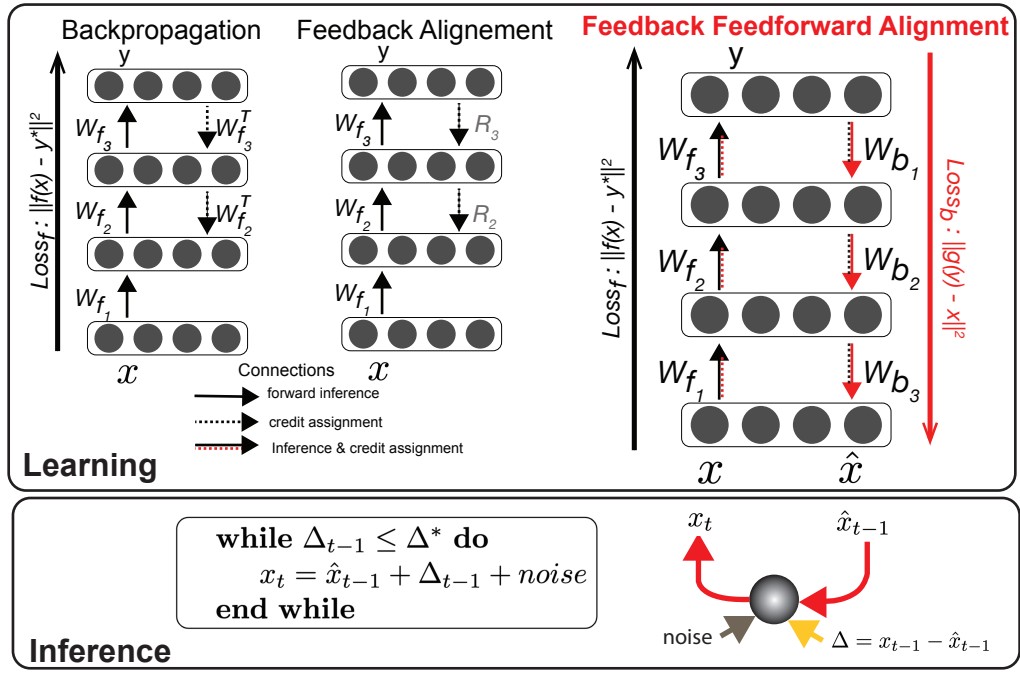

Figure 1: Feedback-Feedforward Alignment. Learning: backpropagation and feedback alignment train a discriminator with symmetric $W_f^T$ or fixed random $R_i$ weights, respectively. FFA maps input $x$ to latents $y$ as in a discriminator but also reconstructs the input $\hat{x}$ from the latent. The forward and backward pathways also pass gradients back for their counterpart performing inference in the opposite direction. Inference: We run forward and feedback connections trained under FFA in a loop to update the activations ($x$) for each of the inference tasks e.g. mental imagery. $\Delta$ shows the difference between the input signal and the reconstructed (output). $*$ shows the imposed upper bound. See algorithm 2 in Section 8.4.

- We suggest and verify that training feedforward and feedback connections for discrimination and reconstruction respectively, induces noise robustness.

- We show that FFA can flexibly support an array of versatile visual inferences such as resolving occlusion, hallucination, and visual imagery.

## 2 Related Work

### 2.1 Models of visual perception and inference in the brain

In going beyond purely feedforward models, there is a large hypothesis space of recurrent neural network models (RNNs), and training inference into an RNN via backpropagation through time raises severe questions about bio-plausibility of the learning algorithm as well as architecture (Lillicrap and Santoro, 2019b). Prior work on RNNs for visual classification whether through complex architecture search (Nayebi et al., 2022) or through imposing theoretically motivated lateral recurrent connections (Tang et al., 2014, 2018) has shown benefits for the classification loss but was not geared to improve our understanding of how feedback or recurrence supports visual inference. On the other hand, there is increasing evidence supporting distinct phases of processing pertaining to perception and inference which parallels the notion of bottom-up versus top-down processing. Recent studies suggest that feedforward and feedback signaling operate through distinct "channels," enabling feedback signals to influence the forward processing without directly affecting the forward-propagated activity, including feedback interaction being more dominant during spontaneous, putatively internally generated activity periods (Semedo et al., 2022; Kreiman and Serre, 2020). Thus, implementing recursion

through feedforward and feedback-dominated phases, as we suggest in FFA, has an anatomical and physiological basis.

---

**Algorithm 1** Training a simple three-layer network by FFA

---

**Learning**
parameters: $W_{f_1}, W_{f_2}, W_{b_1}, W_{b_2}$
**for** $epoch < n\_epochs$ **do**
$\quad\quad Y = W_{f_2} \cdot h_f; h_f = W_{f_1} \cdot X$
$\quad\quad e_f = T - Y$
$\quad\quad Loss_f = \frac{1}{2} e_f^T \cdot e_f$
$\quad\quad \Delta W_{f_2} = -e_f^T \cdot h_f^T, \Delta W_{f_1} = -W_{b_2} \cdot e_f \cdot X^T$ ;    // forward updates
$\quad\quad \hat{X} = W_{b_1} \cdot h_b; h_b = W_{b_2} \cdot Y$
$\quad\quad e_b = X - \hat{X}$
$\quad\quad Loss_b = \frac{1}{2} e_b^T \cdot e_b$
$\quad\quad \Delta W_{b_1} = -e_b^T \cdot h_b^T, \Delta W_{b_2} = -W_{f_1} \cdot e_b \cdot Y^T$ ;    // backward updates
**end**

---

Internally generated perceptual experiences, such as hallucinations, dreams, and mental imagery evoke vivid experiences that mimic the perception of real-world stimuli. Neuroimaging studies demonstrate an overlap in neural activation between internally generated experiences and perception suggesting a shared neural substrate for generating and processing sensory information (Ganis et al., 2004; Pearson, 2019; Pearson et al., 2008; Abid et al., 2016; Dijkstra et al., 2017). While studies have provided insights into the brain regions involved in these phenomena, the neural mechanisms and computations underlying hallucination and imagery remain a topic of ongoing research and debate. One challenge is that hallucinations and imagery are subjective experiences that are difficult to objectively measure and study (Pearson et al., 2008). Additionally, the neural correlates of these experiences can vary across individuals and different types of hallucinations (Suzuki et al., 2017, 2023). Thus, computational modeling of how comparable phenomena can emerge in neural networks, without explicitly training for complex non-bio-plausible generative objective functions (Goodfellow et al., 2014), would help elucidate the neural mechanisms that may underpin these internally-generated perceptions.

## 2.2   Bio-plausible training

Our work also falls within the class of bio-plausible extensions of the original BP algorithm that try to avoid the weight transport problem (Grossberg, 1987). In an effort to mitigate the weight transport problem, (Lillicrap et al., 2016) developed an algorithm known as Feedback Alignment (FA). This algorithm employs random and fixed feedback connections to propagate errors, initiating a line of research that explored different variations of random, yet fixed, feedback connections for credit assignment (Nøkland, 2016; Crafton et al., 2019; Liao et al., 2015). Another line of work uses a strategy that still aims for BP-like symmetric weights while circumventing weight transport by designing a training objective for the feedback path that encourages symmetry (Akrout et al., 2019). For example, augmenting a reconstruction loss with weight decay will constrain solutions to the transpose in the linear setting (Kunin et al., 2019, 2020). However, our method differs in two key ways. First, those methods required invoking a separate gradient pass to train the feedback weights whereas we accomplish the training of feedback with the same feedforward network, thus adding no other hidden paths. Second, those methods explicitly seek symmetry whereas we do not constrain the stage-wise feedback weights, only their end-to-end goal. Our algorithm resembles the stage-wise reconstruction in target propagation (TP) which could also result in end-to-end propagation of latent representations back to inputs if noise at each local propagation step is sufficiently small (Bengio, 2014; Lee et al., 2014). Unlike the layerwise implementation of TP, our training approach is end-to-end, and we do not use any BP training on the penultimate layer of the discriminator.

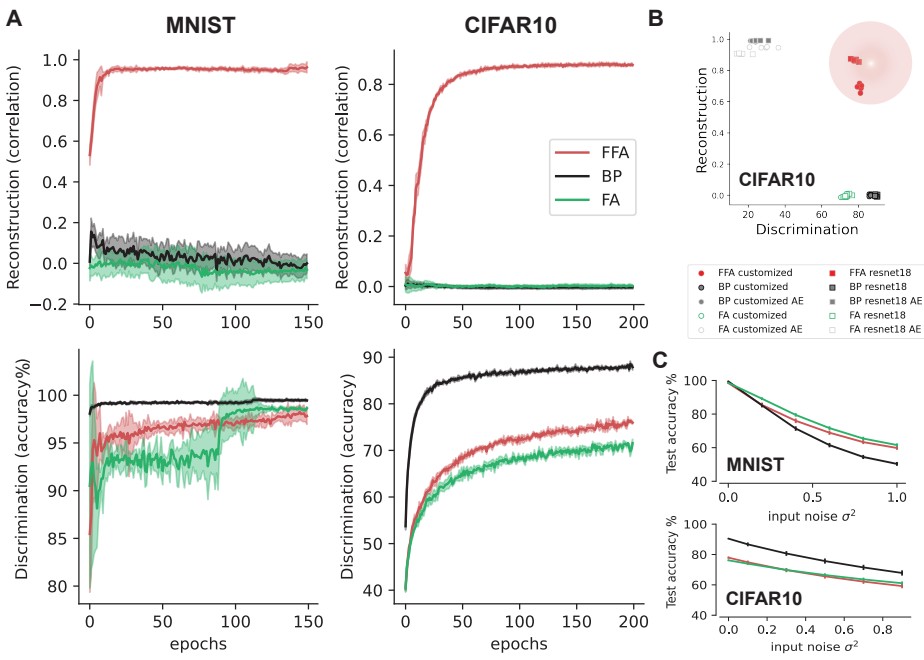

Figure 2: Co-optimization in FFA. A) Accuracy and reconstruction performance for FFA and control algorithms as a function of epochs. B) Dual-task performance for a variety of feedforward discriminative and autoencoder architectures trained under BP or FA compared to FFA training (details for architecture in Suppl. 8.1). The shaded area represents the desired corner. C) Robustness to input Gaussian noise ($\mu = 0$ and varying $\sigma^2$ between 0 and 1) as measured by test accuracy on the noisy input.

## 3 Feedback-Feedforward Alignment

During the training, BP uses a computational graph to backpropagate the error to the hidden layers (Figure 1). This computational graph is a linear neural network that is the transpose of the forward neural network and is constantly updated every time the forward weights are updated. FA and in general the family of the random feedback gradient path such as DFA, use random values and do not update the backward weights during the training. FFA in essence runs two FA algorithms to train the forward pass and backward pass alternatively. The FFA diagram in Figure 1 highlights its two distinguishing features: feedback (decoder) has an end-to-end goal and co-opting of the forward discriminator path (encoder) to train this decoder. A pseudo-code for a simple three-layer neural network to clarify the parameter updates in FFA is in algorithm 1.

Below, we compare how FFA operates on MNIST across two architectures (fully connected and convolutional) and on CIFAR10 using a ResNet architecture by directly reconstructing from the ten-dimensional discriminator output. For details on the architecture please refer to Supplementary material 8.1. For each architecture, we compare FFA to BP and feedback alignment (FA) (Lillicrap et al., 2016) training of a single objective (feedforward discrimination or an autoencoder loss) resulting in 5 control models: FFA, BP, FA, BP-AE, and FA-AE. The purpose of these controls was to verify that the properties of gradient descent on a single loss at the output does not trivially invoke reconstruction of input for example in BP-trained discriminative networks.

### 3.1 FFA achieves the co-optimization of discrimination and reconstruction

We highlight performance results on a convolutional architecture but also report results on a fully connected architecture. Convolutional architectures are potentially of greater interest because they are used for scaling up algorithms to larger datasets. Furthermore, convolutional architectures tend to expose greater performance gaps between BP and FA (Bartunov et al., 2018). FFA-trained networks achieved digit discrimination performance on par with FA but slightly below BP (Figure 2). However, on MNIST, reconstruction performance is exceedingly high. Critically, we were also interested in

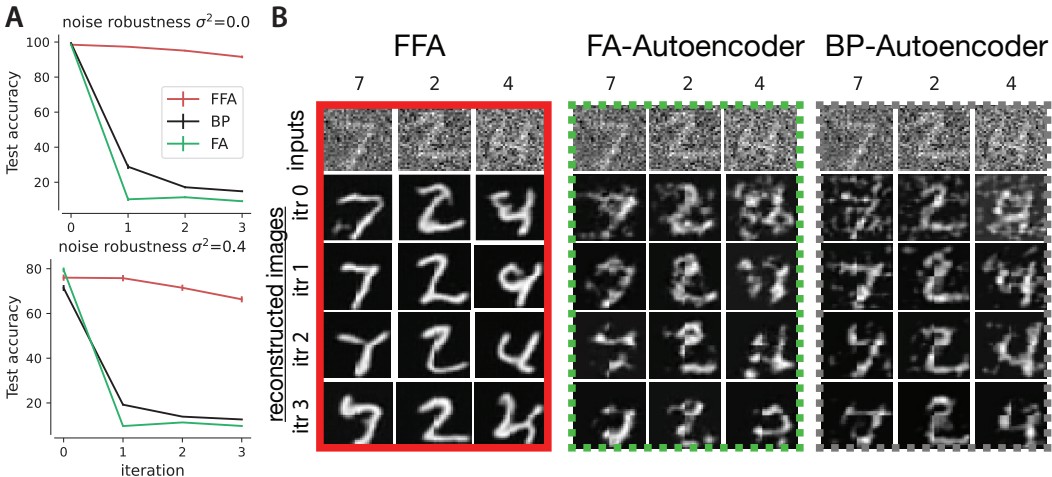

Figure 3: Denoising in FFA. Closed-loop inference on noisy inputs ($\sigma^2 = 0.4$) performed by FFA and control algorithms assuming a static read-out for discrimination set by iteration 0. Shown at right, the sample reconstructions recovered by FFA and control autoencoders over 4 iterations (no clipping or other processing was performed on these images).

seeing if FFA could co-train, using only the discriminator weights for credit assignment, a digit reconstruction path. We found that FFA produced reconstruction on par with a BP-trained autoencoder for convolutional architectures while slightly lagging the autoencoder standard for reconstruction on fully connected architectures. Thus, within the same network, FFA co-optimizes two objectives at levels approaching the high individual standards set by a BP-trained discriminator and a BP-trained reconstructor (see Figure 2 and Figure Suppl. 7). In FFA, like FA, the feedforward and feedback weights aligned over training (Lillicrap et al., 2016), but only in FFA, alignment is useful for reconstruction, presumably because both paths are free to align to each other which breaks the random feedback constraint of FA. In examining discrimination versus reconstruction performance, these can be mutually exclusive. For example, single objective networks tend to improve along one axis or the other. In contrast, FFA-trained networks moved toward the top-right corner of the plot indicating co-optimization along both axes (Figure 2, scatter plot). As shown in Figure 2B for CIFAR10, FFA and FA both struggle to keep up with BP, so for the rest of the paper regarding inference, we focus on MNIST.

### 3.2 FFA induces robustness to image noise and adversarial attacks

Although in FFA training, we did not use any noise augmentation, as we show in this section, the network trained under FFA developed robustness to noise and adversarial attacks relative to the BP control. Previous works showed that BP networks are vulnerable to noise and highlighted that FA-trained networks are surprisingly robust (Goodfellow et al., 2015; Akrout, 2019). When pixel noise was used to degrade input characters, we found that FFA was more robust than BP conferring some of the same robustness seen in FA (Figure 2C). This advantage of FFA and FA over BP was also true for gradient-based white-box adversarial attacks (Figure Suppl. 9).

## 4 Flexible visual inference through recursion

While FFA is not explicitly a recurrent network, by coupling the feedforward and feedback pathways through mutual learning of dual, complementary losses, it may indirectly encourage compatibility in their inference processes. That is, we can run the network in a closed loop, passing $\hat{x}$ from the decoder back in as input to the encoder (replacing $x$) (see Figure 1). In this section, we explore the capabilities of FFA in dealing with missing information (noise or occlusion) and in generation (visual imagery, hallucinations, or dreams). It is worth noting that FFA was not trained to perform any of these tasks and was only trained for discrimination and reconstruction of clean images, conditioned on this discrimination. The inference algorithm we use in this section relies on two main components:

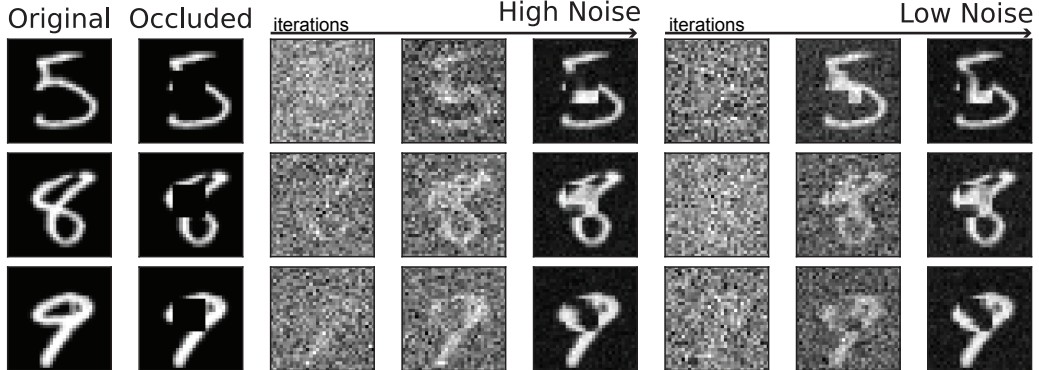

Figure 4: Resolving occlusion. A 15x15 black square occludes the digits in the first columns as shown in the second column. Each row shows a sample occluded digit (5, 8, and 9) and the corresponding resolved images under high noise and low noise conditions. For high noise and low noise visual inference, the resolved digit is depicted in 5th and the last columns, respectively. For details regarding the sample intermediate iterations refer to Figure Suppl. 10.

recursion and noisiness of inference in each recursion. The algorithm was developed in (Kadkhodaie and Simoncelli, 2021) for denoising autoencoders based on *Empirical Bayes Theorem* (Miyasawa, 1961). Although FFA is not trained as a denoiser autoencoder (no noisy input was used during training), we hypothesized that since it exhibits robustness to noise properties, then the theory applies here and the algorithm can be adapted to draw effective inferences from the representation learned by FFA. We especially focused on the effect of the noisiness of inference to inform the computational role of noisy responses in actual biological neurons, as the utility of noisy responses remains largely unknown despite extensive active research (Echeveste and Lengyel, 2018; Findling and Wyart, 2021; McDonnell and Ward, 2011).

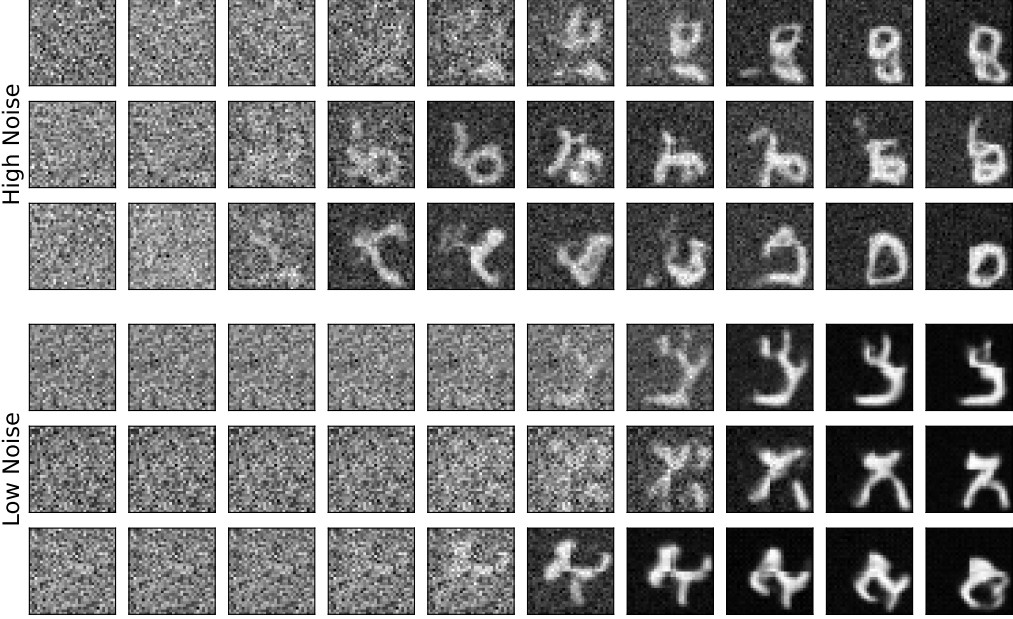

Figure 5: Hallucination. Without external input, we let the inference algorithm run on the FFA-trained network until convergence (the last column) for high noise (upper) and low noise (lower) inference. The sample iterations are linearly spaced and for high noise, there are typically twice as many iterations needed. Refer to Section 8.8 for iteration values.

### 4.1 Denoising

As a first step toward future recurrent processing within FFA, we simply ran the network in a closed loop, passing $\hat{x}$ from the decoder back in as input to the encoder (replacing $x$) (see Figure 1) and found that both discrimination and reconstruction performance is sustained over iterations similar to an autoencoder whereas BP and FA discriminators change over multiple closed-loop iterations – when their feedback path from training was used for generative inference in runtime – and thus would require a dynamic decoder to recover any performance (Figure 3).

### 4.2 Resolving occlusions

We occlude parts of an input image by a blank square and run the network inference. The assumption here is that the occluded image was briefly presented and during the inference, the original image is not accessible throughout inference. Figure 4 shows examples of completion of the pattern using FFA. Even though for high noise inference more iterations were needed, the generated samples do not reflect any superiority compared to low noise inference which took fewer iterations to converge.

### 4.3 Hallucination

Visual hallucinations refer to the experience of perceiving objects or events even when there is no corresponding sensory stimulation that would typically give rise to such perceptions. As mentioned above, the spontaneous activity in V1 is linked to the vividness of hallucinated patterns. Here, we let the FFA-trained network run through the inference algorithm starting from Gaussian noise at the input and adding noise in each iteration. As shown in Figure 5, when in the high noise regime ($\beta = 0.2$), the quality of hallucinated digits is better compared to the low noise regime ($\beta = 0.99$, for the definition of $\beta$ see Section 8.5). Given that the noise in the inference algorithm controls the convergence rate (Kadkhodaie and Simoncelli, 2021), these results suggest that the computational role of spontaneous activity in generating stronger hallucinated percepts may be the refinement of the hallucinated patterns.

### 4.4 Mental imagery

Visual mental imagery refers to the ability to create mental representations or pictures of visual information in the absence of actual sensory input (Pearson et al., 2015; Colombo, 2012). A key distinction between mental imagery and hallucinations is that mental imagery involves *voluntarily* creating mental images through imagination, while hallucinations are involuntary sensory perceptions. To implement the voluntary, top-down activation of a percept (e.g. '9'), we add the average activation pattern of the category in the latent layer to each recursion in the inference algorithm. Presumably, the brain has a recollection of the category which can be read out from memory during mental imagery. Figure 6 shows that as noise in the inference goes higher, so does the quality of the imagined digits.

## 5 Limitations

We acknowledge several limitations of the Feedback-Feedforward Alignment (FFA) framework in its current form. One key limitation is the difficulty of scaling FFA to larger datasets such as ImageNet. While we observed gaps in performance compared to classical backpropagation (BP) on CIFAR10, we found little difference compared to the Feedback Alignment (FA) baseline in discrimination performance on smaller datasets such as CIFAR10. However, it is yet possible that FFA could be more suitable at large scale for specific architectures, such as transformers, where layer sizes do not decrease towards the output layer. Scaling up FFA requires further theoretical and empirical exploration. Another limitation is related to the assessment of the generated inferences. Currently, the evaluation relies primarily on visual inspection. Although we included classifier accuracy reports for denoising, it assumes that perception arises solely from top activations and that bottom hierarchy activation (such as V1) does not directly contribute to perception. Enhancing the evaluation methodology to incorporate more objective measures and quantitative assessments of generated inferences would strengthen the framework. Furthermore, while FFA demonstrates a balance between discrimination performance, efficient learning, and robust recurrent inference, it is important to acknowledge that FFA may not fully capture all aspects of the biological brain. The

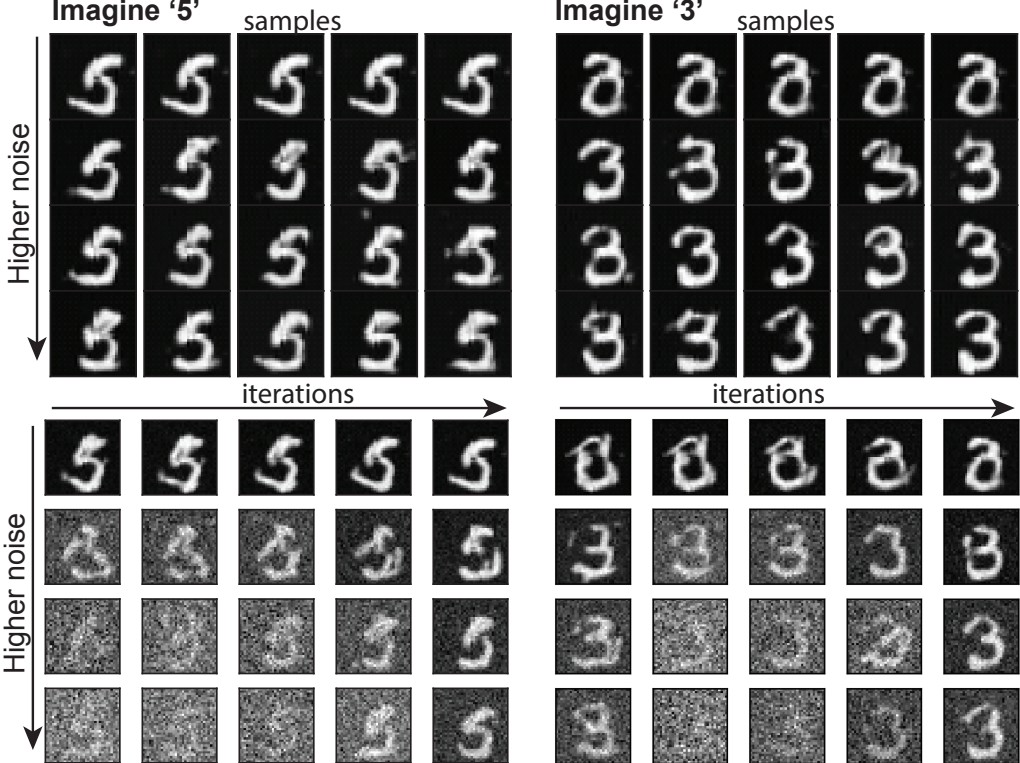

Figure 6: Visual imagery. Generated samples (upper panels) using the inference algorithm on the FFA-trained network when top-down signal '5' (left) and '3' (right) was activated. The sample iterations (equally spaced) for sample generations were shown in the lower panel. Each row corresponds to an inference noise level. Refer to Section 8.7 for iteration and $\beta$ values

framework represents a step towards understanding the brain's mechanisms but may still fall short in faithfully replicating the intricacies of neural processing such as the precise dynamics of neural responses or differences in tuning among feedforward and feedback neural subtypes, though it takes a step toward those directions. Overall, these limitations highlight the need for further research and development to address the scalability of FFA, refine evaluation methodologies, and gain deeper insights into the biological plausibility of the framework. Overcoming these limitations will pave the way for more effective and robust alternatives to BP, advancing the understanding and application of neural network training algorithms to neurobiology.

# 6 Conclusions

In moving beyond classical error backpropagation training of a single-objective, feedforward network, we have presented a feedforward-feedback algorithm that trains neural networks to achieve mutualistic optimization of dual objectives. Co-optimization provides attendant advantages: avoids weight transport, increases robustness to noise and adversarial attack, and gives feedback its own runtime function that allows closed-loop inference. Through our experiments, we demonstrated that the network trained using the FFA approach supports various visual inference tasks.

# 7 Broader Impacts

This work has broader impacts that include advancing our understanding of human perception, enhancing the robustness and performance of neural networks, helping to identify the emergence of closed-loop inference in larger networks for real-time applications, and potential implications for clinical research of mental disorders. By studying the neural mechanisms underlying visual perception, this research contributes to our understanding of natural and artificial vision.

## Acknowledgments

T.T. is supported by NIH 1K99EY035357-01. This work was supported by NIH RF1DA056397, NSF 1707398, Gatsby Charitable Foundation GAT3708, NIH R34 (NS116739), Klingenstein-Simons fellowship, Sloan Foundation fellowship and Grossman-Kavli Scholar Award, as well as a NVIDIA GPU grant, and was performed using the Columbia Zuckerman Axon GPU cluster.

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

# 8 Supplementary

## 8.1 Model architectures

For the experiment on a fully connected architecture, we used a 4-layer network with [1024, 256,256,10] neurons in each layer and ReLU non-linearity between layers. For the experiment on a convolutional architecture, we used a modified version of ResNet (He et al., 2015), where the last convolutional layer has the same number of channels as classes, and an adaptive average pooling operator is used to read out of each channel (see below). Since the last layer doesn't have any learnable parameters, the penultimate layer can be as large as desired which works fine for FFA. The convolutional architecture consists of 11 convolutional layers with 658,900 trainable parameters in total. For autoencoder controls (trained under BP or FA), we additionally trained a linear decoder on the activations of the penultimate layer to assess the linear separability of the representation learned by autoencoders.

Code is available at: https://github.com/toosi/Feedback_Feedforward_Alignment

## 8.2 More control networks: autoencoders

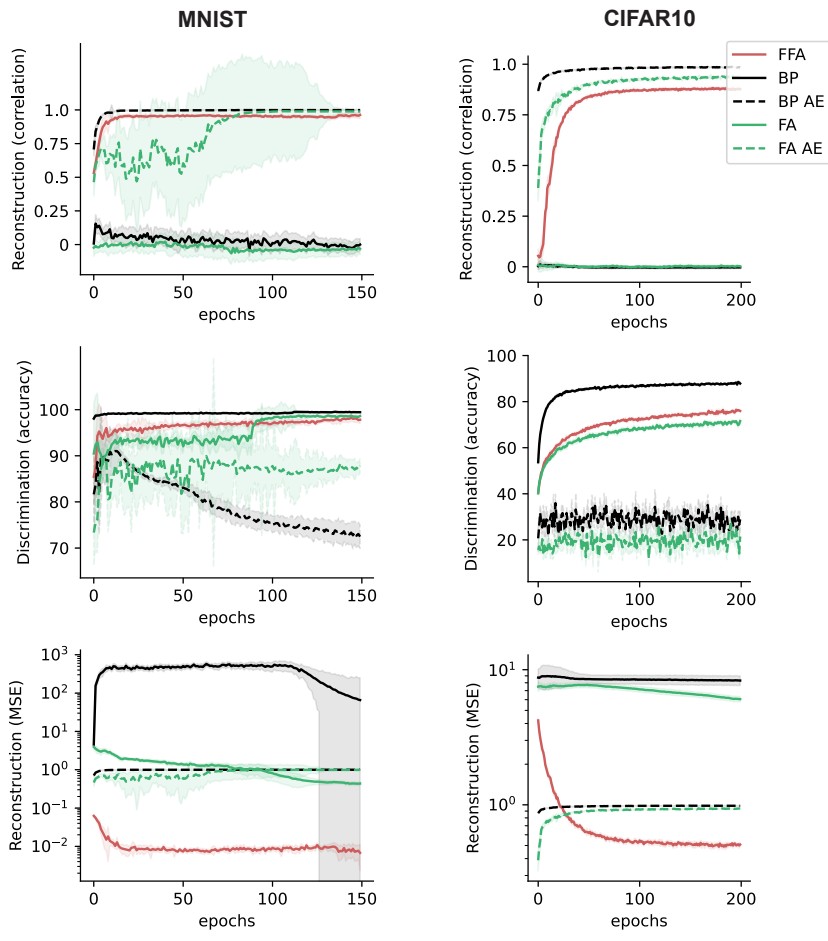

Figure 7: Co-optimization in FFA compared to single objective (either discrimination or reconstruction) control networks for MNIST and CIFAR10 (extensive version of Figure 2).

## 8.3 Alignment between feedback and feedforward

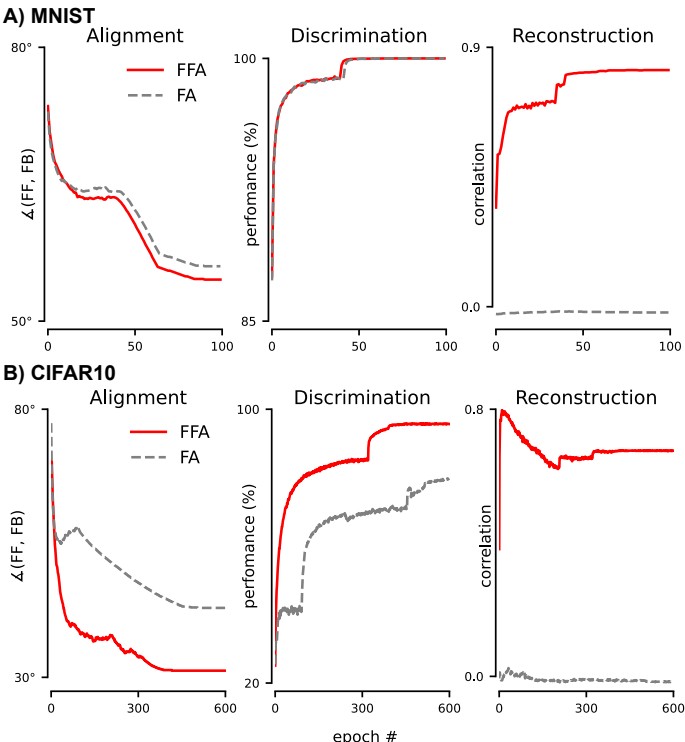

Figure 8: Alignment of feedback (FB) and feedforward (FF) weights for A) MNIST B) CIFAR10 datasets over the course of training shown in the first column. The smaller the angle between the FF and FB weights, the higher the alignment. The second and third columns show discrimination and reconstruction performances on test sets.

## 8.4 Robustness assessment

We added Gaussian noise with zero mean and varied the variance $\sigma^2 = [0.0, 0.2, 0.4, 0.8, 1.0]$ to assess the robustness of models to input noise. We also performed a widely used white box adversarial attack Fast Gradient Sign Method (FGSM) (Goodfellow et al., 2015). FGSM can be summarized by

$$x^{'} = x + \epsilon sign(\Delta_x J(x, y^*))$$

where $\sigma$ is the magnitude of the perturbation, $J$ is the loss function and $y^*$ is the label of $x$. While in BP this perturbation is computed through transposed forward parameters, for FFA and FA, we used their gradient pass parameters which are learned feedback and random feedback, respectively. We used a range of $\epsilon$ to cover the interval between 0 to 1.

## 8.5 Visual inference algorithm

We adapt the sampling algorithm developed in (Kadkhodaie and Simoncelli, 2021) to implement the visual inference in FFA-trained networks. $\beta$ parameter which varies between 0 and 1 controls the proportion of injected noise ($\beta = 1$ indicates no noise). The injected noise together with the gradient step size controls the number of iterations for convergence.

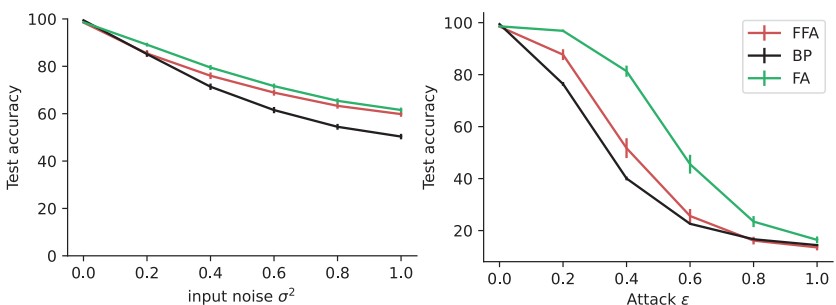

Figure 9: Robustness to Gaussian noise and adversarial attacks for MNIST. Robustness to noise and adversarial attacks in input (image) space for FFA and control algorithms. FA and FFA both exhibit more robustness than BP-trained discriminators.

---

**Algorithm 2** The core visual inference algorithm
(adapted with minimal modification from Kadkhodaie and Simoncelli (2021)))

---

parameters: $\sigma_0, \sigma_L, h_0, \beta$
initialization: $t = 1$, draw $x_0 \sim \mathcal{N}(0.5, \sigma_0^2 I)$
$\mathscr{F}$: trained feedforward
$\mathscr{B}$: trained feedback
**while** $\sigma_{t-1} \leq \sigma_L$ **do**
$\quad h_t = \frac{h_0 t}{1 + h_0(t-1)}$
$\quad \hat{x}_{-1} = \mathscr{B}(\mathscr{F}(x_{t-1}))$
$\quad d_t = x_{t-1} - \hat{x}_{-1}$
$\quad \sigma_t^2 = \frac{\|d_t\|^2}{N}$
$\quad \gamma_t^2 = \left((1 - \beta h_t)^2 - (1 - h_t)^2\right) \sigma_t^2$
$\quad$ Draw $z_t \sim \mathcal{N}(0, I)$
$\quad x_t \leftarrow x_{t-1} + h_t d_t + \gamma_t z_t$
$\quad t \leftarrow t + 1$
**end**

---

## 8.6 Visual occlusion

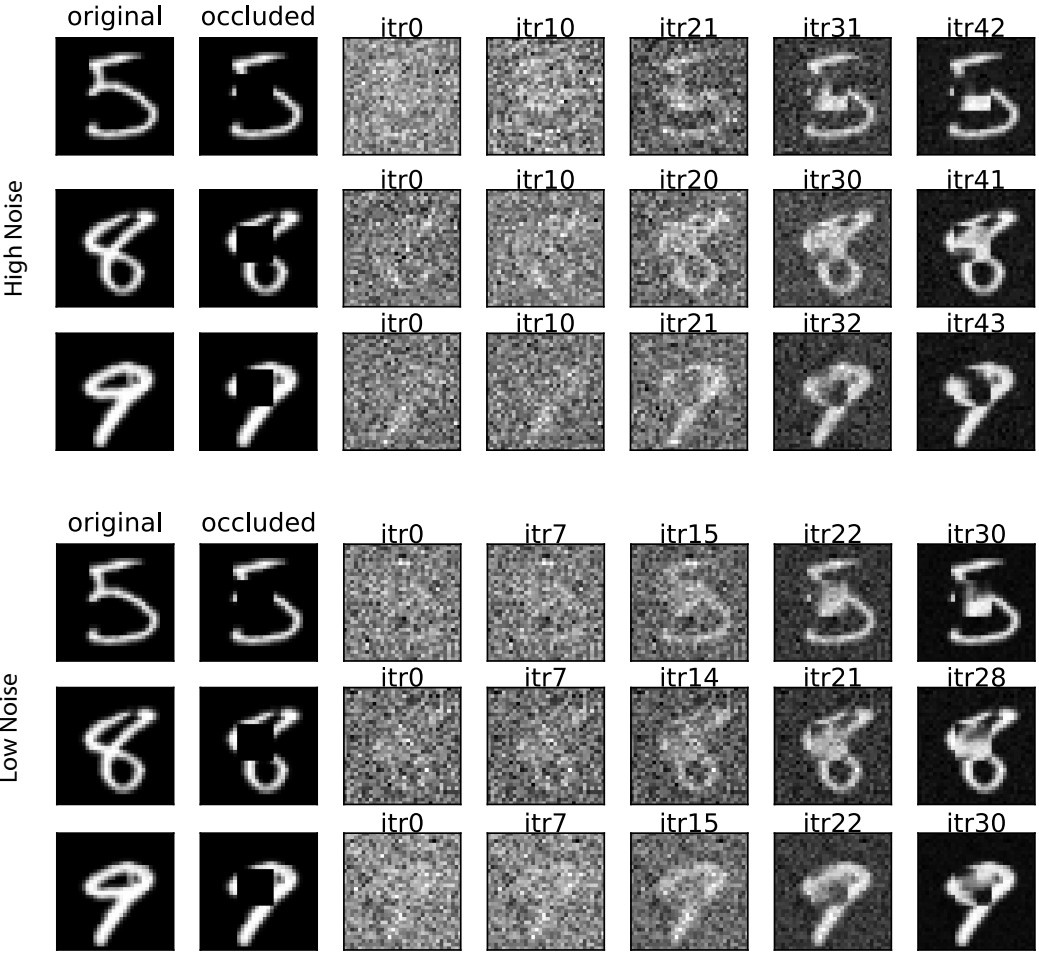

Figure 10: Visual occlusion related to Figure 4 in the main text.

## 8.7 Visual imagery

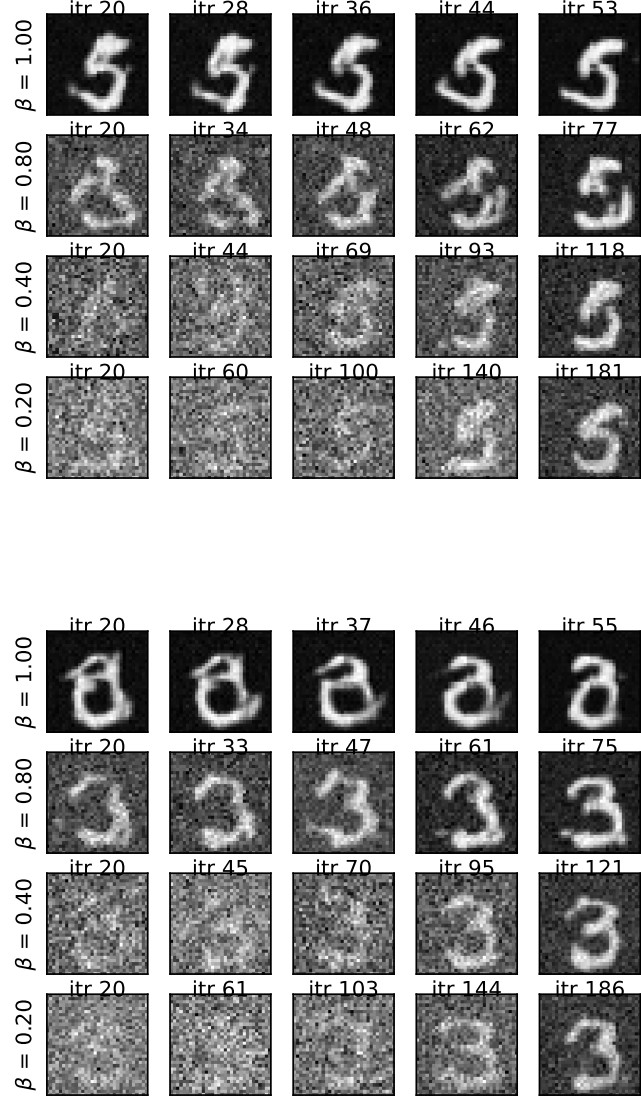

Figure 11: Sample visual imagery related to Figure 6 in the main text.

## 8.8 Hallucinations

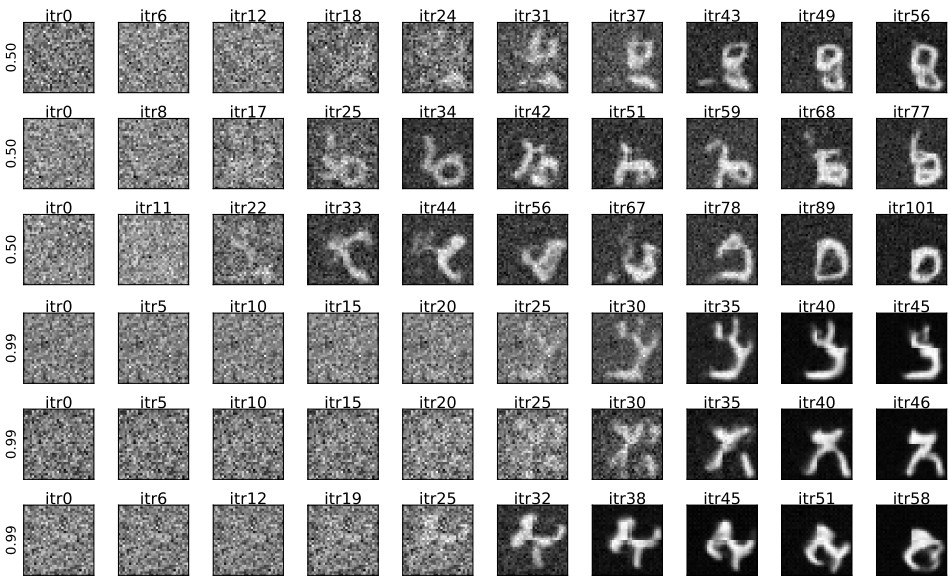

Figure 12: Sample hallucinations related to Figure 5 in the main text.

