```
modelF: DataParallel(
  (module): AsymResLNet10F(
    (conv1): AsymmetricFeedbackConv2d(1, 64, kernel_size=(7, 7),
    stride=(2, 2), padding=(3, 3), bias=False)
    (bn1): BatchNorm2d(64, eps=1e-05, momentum=0.1,
    affine=True, track_running_stats=False)
    (relu): ReLU(inplace=True)
    (conv11): AsymmetricFeedbackConv2d(64, 64, kernel_size=(3, 3),
    stride=(1, 1), padding=(1, 1), bias=False)
    (bn11): BatchNorm2d(64, eps=1e-05, momentum=0.1,
    affine=True, track_running_stats=False)
    (conv12): AsymmetricFeedbackConv2d(64, 64, kernel_size=(3, 3),
    stride=(1, 1), padding=(1, 1), bias=False)
    (bn12): BatchNorm2d(64, eps=1e-05, momentum=0.1,
    affine=True, track_running_stats=False)
    (conv21): AsymmetricFeedbackConv2d(64, 64, kernel_size=(3, 3),
    stride=(1, 1), padding=(1, 1), bias=False)
    (bn21): BatchNorm2d(64, eps=1e-05, momentum=0.1,
    affine=True, track_running_stats=False)
    (conv22): AsymmetricFeedbackConv2d(64, 128,
    kernel_size=(3, 3),
    stride=(1, 1), padding=(1, 1), bias=False)
    (bn22): BatchNorm2d(128, eps=1e-05, momentum=0.1,
    affine=True, track_running_stats=False)
    (downsample1): AsymmetricFeedbackConv2d(64, 128,
    kernel_size=(1, 1), stride=(1, 1), bias=False)
    (bn23): BatchNorm2d(128, eps=1e-05, momentum=0.1,
    affine=True, track_running_stats=False)
    (conv31): AsymmetricFeedbackConv2d(128, 128,
    kernel_size=(3, 3),
    stride=(2, 2), padding=(1, 1), bias=False)
    (bn31): BatchNorm2d(128, eps=1e-05, momentum=0.1,
    affine=True, track_running_stats=False)
    (conv32): AsymmetricFeedbackConv2d(128, 128,
    kernel_size=(3, 3),
    stride=(1, 1), padding=(1, 1), bias=False)
    (bn32): BatchNorm2d(128, eps=1e-05, momentum=0.1,
    affine=True, track_running_stats=False)
    (conv41): AsymmetricFeedbackConv2d(128, 128,
    kernel_size=(3, 3),
    stride=(1, 1), padding=(1, 1), bias=False)
    (bn41): BatchNorm2d(128, eps=1e-05, momentum=0.1,
    affine=True, track_running_stats=False)
```

```
436    (conv42): AsymmetricFeedbackConv2d(128, 10,
437    kernel_size=(3, 3),
438    stride=(1, 1), padding=(1, 1), bias=False)
439    (bn42): BatchNorm2d(10, eps=1e-05, momentum=0.1,
440    affine=True, track_running_stats=False)
441    (downsample2): AsymmetricFeedbackConv2d(128, 10,
442    kernel_size=(1, 1), stride=(2, 2), bias=False)
443    (avgpool): AdaptiveAvgPool2d(output_size=(1, 1)))
444  )
445
446  modelB: DataParallel(
447    (module): AsymResLNet10B(
448      (upsample2): AsymmetricFeedbackConvTranspose2d(10, 128,
449      kernel_size=(1, 1), stride=(2, 2), output_padding=(1, 1),
450      bias=False)
451      (bn42): BatchNorm2d(10, eps=1e-05, momentum=0.1,
452      affine=True, track_running_stats=False)
453      (conv42): AsymmetricFeedbackConvTranspose2d(10, 128,
454      kernel_size=(3, 3), stride=(1, 1), padding=(1, 1), bias=False)
455      (relu): ReLU(inplace=True)
456      (bn41): BatchNorm2d(128, eps=1e-05, momentum=0.1,
457      affine=True, track_running_stats=False)
458      (conv41): AsymmetricFeedbackConvTranspose2d(128, 128,
459      kernel_size=(3, 3), stride=(1, 1), padding=(1, 1), bias=False)
460      (bn32): BatchNorm2d(128, eps=1e-05, momentum=0.1,
461      affine=True, track_running_stats=False)
462      (conv32): AsymmetricFeedbackConvTranspose2d(128, 128,
463      kernel_size=(3, 3), stride=(1, 1), padding=(1, 1), bias=False)
464      (bn31): BatchNorm2d(128, eps=1e-05, momentum=0.1,
465      affine=True, track_running_stats=False)
466      (conv31): AsymmetricFeedbackConvTranspose2d(128, 128,
467      kernel_size=(3, 3), stride=(2, 2), padding=(1, 1),
468      output_padding=(1, 1), bias=False)
469      (bn23): BatchNorm2d(128, eps=1e-05, momentum=0.1,
470      affine=True, track_running_stats=False)
471      (upsample1): AsymmetricFeedbackConvTranspose2d(128, 64,
472      kernel_size=(1, 1), stride=(1, 1), bias=False)
473      (bn22): BatchNorm2d(128, eps=1e-05, momentum=0.1,
474      affine=True, track_running_stats=False)
475      (conv22): AsymmetricFeedbackConvTranspose2d(128, 64,
476      kernel_size=(3, 3), stride=(1, 1), padding=(1, 1), bias=False)
477      (bn21): BatchNorm2d(64, eps=1e-05, momentum=0.1,
478      affine=True, track_running_stats=False)
479      (conv21): AsymmetricFeedbackConvTranspose2d(64, 64,
480      kernel_size=(3, 3), stride=(1, 1), padding=(1, 1), bias=False)
481      (bn12): BatchNorm2d(64, eps=1e-05, momentum=0.1,
482      affine=True, track_running_stats=False)
483      (conv12): AsymmetricFeedbackConvTranspose2d(64, 64,
484      kernel_size=(3, 3), stride=(1, 1), padding=(1, 1), bias=False)
485      (bn11): BatchNorm2d(64, eps=1e-05, momentum=0.1,
486      affine=True, track_running_stats=False)
487      (conv11): AsymmetricFeedbackConvTranspose2d(64, 64,
488      kernel_size=(3, 3), stride=(1, 1), padding=(1, 1), bias=False)
489      (bn1): BatchNorm2d(64, eps=1e-05, momentum=0.1,
490      affine=True, track_running_stats=False)
491      (conv1): AsymmetricFeedbackConvTranspose2d(64, 1,
492      kernel_size=(7, 7), stride=(2, 2), padding=(2, 2),
493      output_padding=(1, 1), bias=False))
494  )
```

 ## 8.2 More control networks: autoencoders

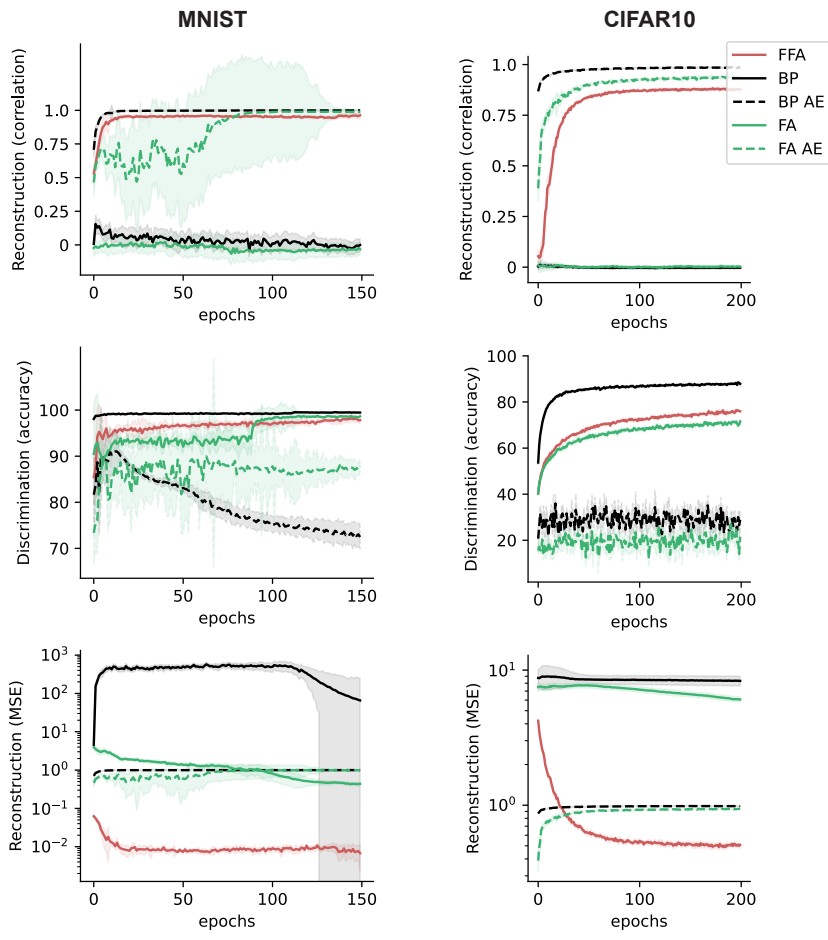

Figure 7: Co-optimization in FFA compared to single objective (either discrimination or reconstruction) control networks for MNIST and CIFAR10 (extensive version of Figure 2).

### 8.3 Robustness assessment

We added Gaussian noise with zero mean and varied the variance $\sigma^2 = [0.0, 0.2, 0.4, 0.8, 1.0]$ to assess the robustness of models to input noise. We also performed a widely used white box adversarial attack Fast Gradient Sign Method (FGSM) (Goodfellow et al., 2015). FGSM can be summarized by

$$x^{'} = x + \sigma sign(\Delta_x J(x, y^*))$$

where $\sigma$ is the magnitude of the perturbation, $J$ is the loss function and $y^*$is the label of $x$. While in BP this perturbation is computed through transposed forward parameters, for FFA and FA, we use their gradient pass parameters which are learned feedback and random feedback, respectively. We used a range of $\epsilon$ to cover the interval between 0 to 1.

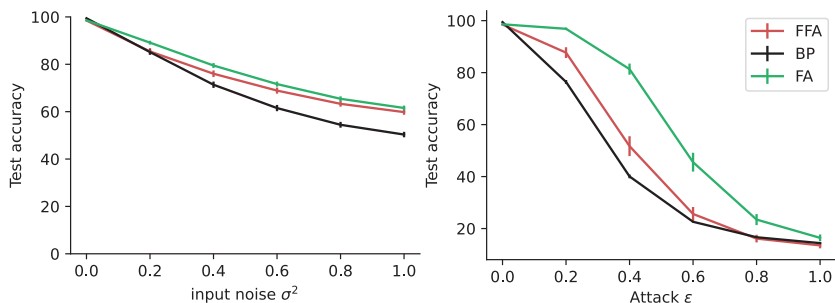

Figure 8: Robustness to Gaussian noise and adversarial attacks for MNIST. Robustness to noise and adversarial attacks in input (image) space for FFA and control algorithms. FA and FFA both exhibit more robustness than BP-trained discriminators.

### 8.4 Visual inference algorithm

We adapt the sampling algorithm developed in (Kadkhodaie and Simoncelli, 2021) to implement the visual inference in FFA-trained networks. $\beta$ parameter which varies between 0 and 1 controls the proportion of injected noise ($\beta = 1$ indicates no noise).

---

**Algorithm 1** *

parameters: $\sigma_0$, $\sigma_L$, $h_0$, $\beta$
initialization: $t = 1$,  draw $x_0 \sim \mathcal{N}(0.5, \sigma_0^2 I)$
**while** $\sigma_{t-1} \leq \sigma_L$ **do**

$\quad h_t = \frac{h_0 t}{1 + h_0(t-1)}$
$\quad d_t = x_{t-1} - \hat{x}_{t-1}$
$\quad \sigma_t^2 = \frac{||d_t||^2}{N}$
$\quad \gamma_t^2 = \left((1 - \beta h_t)^2 - (1 - h_t)^2\right) \sigma_t^2$
$\quad$Draw $z_t \sim \mathcal{N}(0, I)$
$\quad x_t \leftarrow x_{t-1} + h_t d_t + \gamma_t z_t$
$\quad t \leftarrow t + 1$

**end**
Stochastic gradient ascent method for sampling from the implicit prior in a denoiser autoencoder as in Kadkhodaie and Simoncelli (2021)

---

 **8.5  Visual imagery**

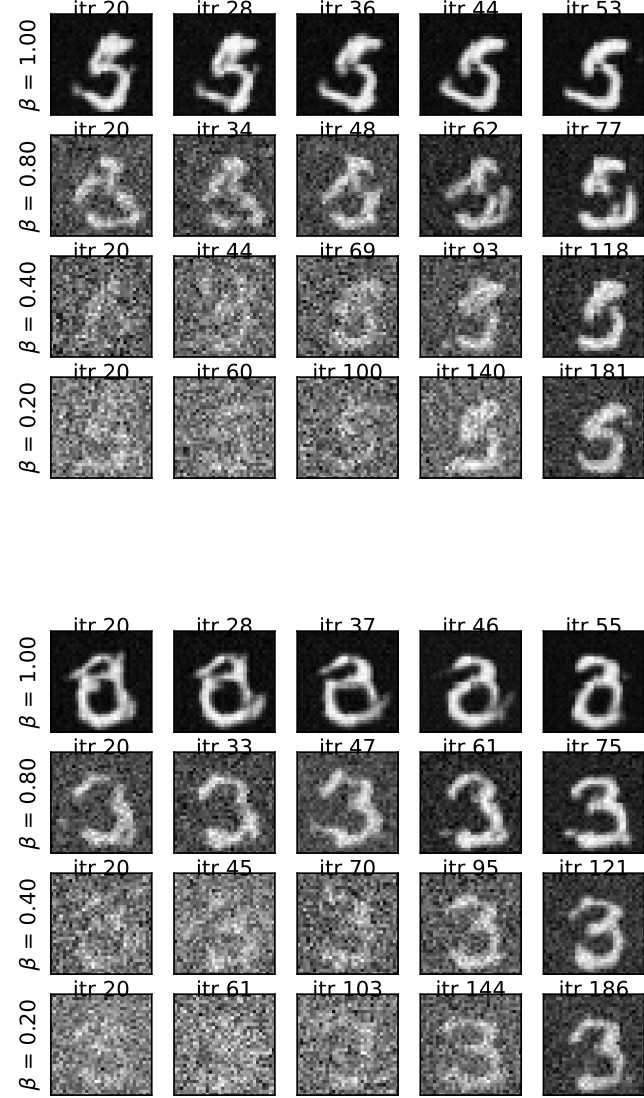

Figure 9: Sample visual imagery related to Figure 6 in the main text.

 ## 8.6  Hallucinations

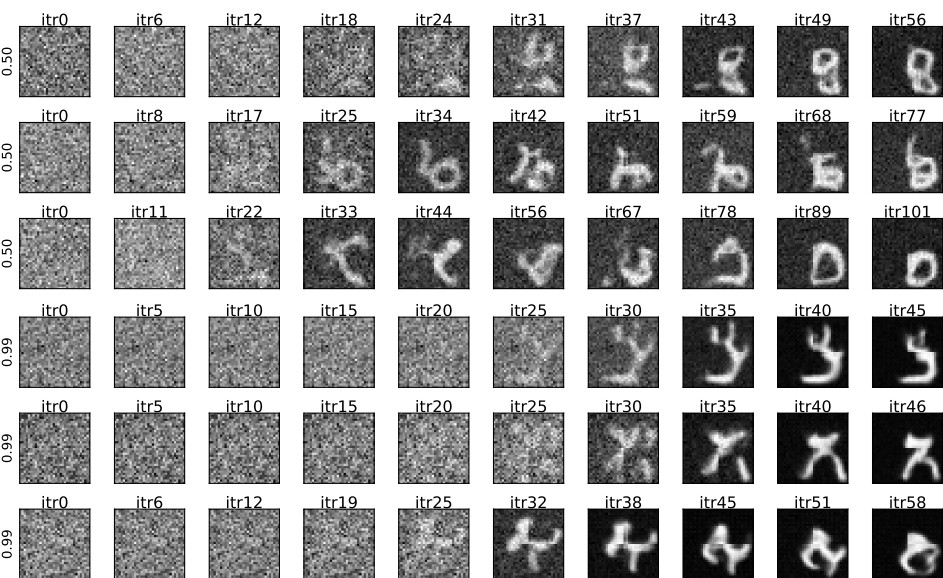

Figure 10: Sample hallucinations related to Figure 5 in the main text.