# OpenReview forum: "Brain-like Flexible Visual Inference by Harnessing Feedback Feedforward Alignment"
_NeurIPS.cc/2023/Conference — NeurIPS 2023 poster_

### Official Review · Reviewer_WNmV · 2023-06-29

**Soundness:** 2 fair
**Presentation:** 3 good
**Contribution:** 3 good
**Rating:** 4
**Confidence:** 3

**Summary:**

The authors present a novel artificial neural network architecture which they relate to visual processing in the brain. In particular, the authors avoid the weight transport problem by proposing that feedback weights can be learned by optimising them to reconstruct the original input (autoencoder). The authors test the model on image datasets and show that the network can both discriminate and reconstruct the original image, as well as replicating biological visual phenomena with respect to resolving occlusions, hallucinations and imagination.

**Strengths:**

- the paper is generally well written
- The proposed model architecture/learning rules are interesting and creative


**Weaknesses:**

- there is a distinct lack of specificity in the formal model description. Its illustration and algorithm is helpful, but I think the clarity of its exact computations would significantly benefit from equations
- The authors do not address analytical aspects of their model. What types of solution should one expect in weight space compared to discrimination/auto-encoder only models? What is the extent of alignment compared to fixed random weights (feedback alignment). The authors write that "co-optimizing discrimination and reconstruction objectives can lead to a mutualistic symbiotic interaction", would it be possible to quantify this? Relately, are there any necessary conditions for this architecture to work, or where it'll break? for example, feedback alignment struggles with harder tasks, should one expect this with FFA?
- The functional benefit of FFA is not very clear to me. For example, backprop still outperforms models even with noise on harder CIFAR task (fig 2), and why are several iterations necessary for discrimination if performance gets worse anyway (fig 3)? Is it not simpler just to train with backprop using two loss functions (discrimination and reconstruction), or is the idea that that's less bio-plausible?
- I'm a little confused about the bio-plausibility of FFA: If I understand correctly, the forward weights W_f affect the activity in the post-synaptic neuron and this activity propagates forward, but the same activity doesn't propogate it the feedback pathway? How is this possible biologically?

**Questions:**

- are there no other computational models of top-down architectures to compare to? The authors cite works in experimental neuroscience which support top-down feedback during visual processing, do other explicit implementations not exist? e.g. are bidirectional variational autoencoders not related? (e.g. Kingma et al. Neurips 2016)
- in the fig 1 inference box, should it not be delta_{t-1} >= delta*? Same for Algorithm 1. I'm not sure the figure caption text '* shows the desired value' is the right language, isn't delta* rather an imposed upper bound which we want to fall below?
- Fig 1 caption last sentence: add full stop
- line 99: "Prior work on RNNs..was not geared to improve our understanding of how feedback or recurrence supports inference of visual details": Could the authors clarify what is meant by 'visual details'?
- For discrimination performance figures percentage signs should be added
- Fig 2B: 'customised' is unclear. Are these fully connected feedforward nets?
- line 132: 'Unlike the original TP, we do not constrain the intermediate stages'. What constraints are the authors referring to?
- line 137: FA is used before it has been defined as feedback alignment
- line 156: 'on MNIST, discrimination performance is exceedingly high'. What is meant here?
- line 162: 'In FFA, like FA, the feedforward and feedback weights aligned over training'. Could the authors show this?
- I advise the authors to use consistent references to supplementary figures (cf. line 162 vs 178). Moreover, remove the space between the figure number and subpanel (e.g. Figure 2C in line 177)
- line 182: z has not been defined. Is this the random sample as in algorithm 1?
- Figure 4: the meaning of the left-to-right in this selection of images is unclear. Is each x position a different iteration? which iteration? the difference between high/low noise also appears very small. The authors argue that low noise "took fewer iterations to converge" but I do not find it very clear from the figure (except the second shown iteration for the 5 example). Similarly, in Figure 6 the authors state that "Figure 6 shows that as noise in the inference goes higher, so does the quality of the imagined digits": this appears true for the 3 digit but not 5. Is there a way to quantify the quality over all digits?
- In section 8.1 the model architecturers are not very readable. A schematic on even textual description would be easier to intepret.
- Equation in section 8.3: is it not supposed to be epsilon in the equation, not sigma


**Limitations:**

The authors do address some limitations, though as I say above I think it should also be highlighted in the paper the lack of theoretical study/support

---

> ### Author Rebuttal · Authors · 2023-08-08
>
> We appreciated the detailed evaluation of the reviewer and their comments to improve the manuscript.
>
> > "Weaknesses:”
>
> We now added a pseudocode to the paper (see general response).
>
> > "The authors ... “
>
> In the general response, we outline the theoretical motivation behind the work and the limitations in theoretical understanding regarding the learning algorithms.
>
> > "What is ...”
>
> We included the alignment analysis to the general response pdf.  Quantification of the mutualistic alignment between forward and backward means they can work with each other’s generation. In Fig.3, using iterative inference, we showed that FFA Feedforward and feedback together induce noise robustness, although neither of the loss functions for these two pathways were trained to be robust to noise.
>
>
> > " Relately, ...”
>
>
> As mentioned in the limitation section of the work, FFA and in essence FA, both have problems with more tapered architectures, that is the architectures where the dimension of representation reduces drastically over depth. In the original FA paper (Lillicrap et al., 2016), they mentioned FA specifically had trouble training an autoencoder which might be related to reconstruction loss and bottleneck nature of the architecture, leading to less degrees of freedom for alignment in the solution space.
>
> > "The functional ...”
>
> We apologize for the lack of clarity regarding the functional benefit of FFA. BP is indeed superior in performance, however, as we mentioned in the Introduction, the implementation of BP is not bio-plausible because it needs symmetric weights in feedforward and feedback pass. In the experiments regarding performance of FFA in discrimination and reconstruction, we included BP as a control to gauge the performance of FFA in comparison with the most performant but bio-implausible algorithm.
>
> > "I'm ...”
>
> We have included the pseudocode for FFA to clarify. When updating the feedback pass weights, the activity that propagated through the forward weights all the way to the top, propagates through the feedback weights all the way down to generate activity the same size of the image (for reconstruction). Thus, forward and backward pathways are distinct, just as there is evidence from feedforward and feedback streams in different lamina in the visual cortex  (Markov et al., 2013; Rockland, 2022).
>
> > "Ques...”
>
> In the literature for bio-plausible learning algorithms (Goudarzi, 2017/2017; Liao et al., 2016; Lillicrap et al., 2016; Meulemans et al., 2020; Moskovitz et al., 2018; Nøkland, 2016), it’s common practice to compare learning algorithms on the same feedforward architecture. There are certainly computational models of top-down feedback as we mentioned in the introduction (and now we added more), however, virtually all of them train the top-down connections using classical error backpropagation. Moreover, to compare the effectiveness of learning, the identical architecture should be used when comparing various training algorithms. Rather, other top-down models such as bidirectional variational autoencoders, have both architectural (additional encoder) and loss function (probabilistic component) complexities on top of a bio-implausible learning algorithm (BP). We would like to clarify that we didn’t aim to model the visual inference properties of feedback connections, rather, these properties fell out of a bio-plausible implementation of our learning algorithm (FFA) that does not rely on a weight symmetry constraint.
>
>
> > "in fig 1...”
>
> The reviewer is correct in all those instances, we fixed the figure and the caption. Thank you.
>
> > " line 99: ...”
>
> We meant the visual ambiguities and occlusions or lack of resolution but we agree that was not evident from what we wrote. We now changed it to “ ..was not geared to improve our understanding of how feedback or recurrence supports visual inference outside of output discrimination performance.”
>
> > "For discri...”
>
> Fixed now, thank you.
>
> > "Fig 2B ...”
>
> Yes, by customized architecture we meant fully-conn. as opposed to the resnet-like conv. architecture as we explained in supplementary 8.1. We now modified the caption of Fig. 2 for clarity.
>
> > "l. 132...?”
>
> Here, we referred to the layer-wise nature of target propagation in contrast to the end-to-end approach in FFA. But we acknowledge that this was not readily clear from what we wrote. We now modified it to: Unlike the layerwise implementation of TP, our training approach is end-to-end.
>
> > "l. 137...”
>
> Fixed it, thank you.
>
> > "l. 156: ...”
>
> There was a typo and we meant reconstruction performance was much higher.
>
> > "l. 162..."
>
> We now added the alignment between feedforward and feedback weights (see pdf above).
>
> > "I advise ...”
>
> Thank you for  reminding us to correct these points to improve the readability of the paper. Now fixed.
>
> > "line 182...”
>
> Thanks for catching the mistake, it was left from an old notation. We now change to \hat{x} as in Fig.1.
>
> > "Fig 4: ...”
>
> We apologize for the lack of clarity in the Fig. 4 and the caption. Please see Fig. 2 in pdf above. We now added the number of iterations to the text for further clarity. Regarding the Fig. 6 statement on the quality of the imagined digits, quantification of the imagined digits would be very beneficial. However, because there is no ground truth, the ultimate gold standard would be human judgment and we would need to ask human subjects to rate the quality of the imagined digits. However, similar to many other studies on generative models, we relied on providing the generated samples here for the reader.
>
> > "In section 8.1 ...”
>
> The model architectures were added to the suppl. for full transparency. We now added more textual description to sec. 8.1 to make the code easier to interpret.
>
> > "Equa ...”
>
> Thank you for pointing out this typo.
>
> > "Limitations”
>
> We added theoretical motivation to the main text (see general response above) and have also added difficulties related to the theory to the Limitation section.

---

> > ### Comment · Reviewer_WNmV · 2023-08-13
> >
> > Thank you to the authors for their detailed response.
> >
> > I still find the proposed model architecture/learning rules as interesting and creative, and would be confident that this would be of value particularly to the neuroscience community. Overall, my issue remains that at is it (even with the promised modifications), the paper falls somewhat short in rigorously demonstrating the benefits of FFA within the high standards set at neurips. I don't think this is helped, as Reviewer 3rRy comments, by the way the model is presented more as an ML model rather than a biological model, and would encourage a more direct relation/comparison to neural/behavioural data in vivo for the future.
> >
> > With the improvements made to the paper, I will slightly increase my contribution score and overall score.

---

### Official Review · Reviewer_3rRy · 2023-06-30

**Soundness:** 2 fair
**Presentation:** 2 fair
**Contribution:** 2 fair
**Rating:** 4
**Confidence:** 4

**Summary:**

This article proposes and exploits a bio-plausible mechanism inspired from the Feedback Alignement : the Feedfback-Feedforward Alignement (FFA). Through a series of experiments (classification, robustness, denoising, generation…) the authors demonstrate that the FFA could be used for various applications.

**Strengths:**

I have found this paper original, as it proposes a solution that solves the weight transport problem in biology without imposing additional constraints such as weight symmetry. I have also found interesting the versatility of the FFA as well as recursion-like inference.

**Weaknesses:**

Despite the very interesting topic and idea, I was not convinced by the experimental parts (see more details below). The main point of this article is the  bio-plausibility of the method: this is a good motivation, but not a result in itself. I would have expected more results in line with biology/psychophysics (see Major Concern 2 ), to show that the proposed algorithms is a good model of the human cognition. Overall the article presents a series of experiments that lacks comparison with 1) similar model in machine learning (see Major Concern 1) and 2) biological data.

Major Concerns:
1) The literature review is weak, especially in the field of machine learning. Plenty of algorithms have already been proposed to model human cognition using feedforward and feedback connections. Some of these works should have served as a benchmark in the proposed experiments. Here are more details:
[1] proposed a deep-learning friendly implementation of the predictive coding algorithms (using backdrop). They are demonstrating that the prediction accuracy gets refined through time. It seems that the model you proposed has lowered accuracy through time (see Fig 3 A : the red curve tends to decrease whatever the noise). How do you explain that ? what’s the point of degrading performance through time ?
[2] proposes a similar network to [1] but more in line with bio-plausible training. It shows refined denoising abilities thought time. How do you compare the FFA with those models ?
[3], [4] have developed VAE-models based on predictive coding. Those are showing nice ‘hallucination’ and denoisiing abilities as well as good classification accuracy in challenging environment (noisy image…). Such models might serve as a good baseline to compare with.
[5], [6] have published numerous articles to show that backprogation could be approximated (in a biology-friendly way) using the predictive coding model. The authors don’t even mention those works !
[7] : have proposed a predictive coding model doing discrimination and classification in the same time. Such a network would have served as a good comparison for the FFA

2) The main interest of the FFA seems to be its bio-plausibility, but the authors keep evaluating it as a machine learning model rather than a biological model. Most of the experiments the authors are showcasing actually show the weaknesses of the model rather than its advantage. Here are some exemples :
* Fig 1 shows that FFA is not as good as a classical Autoencoder to reconstruct, and not as good as a BP feedforward model to classify… The only argument in favor of a FFA is that it could do both in the same time… but with lower performance… this is not very appealing (even more when we look at the CIFAR10 curve).
* Fig 1c shows that FFA is not as good as FA for classification of noisy MNIST and not as good as the BP model for classifying noisy images..
* Fig 3 shows that the more you let the iterative process going, the more the performance degrade… Then what’s the point oh having an iterative inference process ?


I would suggest the authors to modify the way they are evaluating the model. For example why not testing if the FFA is sensitive to perceptual illusion (as in [8]), are why not comparing the number of iterative inference steps with the human reaction time ? Those comparison would tend to show that the FFA is a good model of human cognition.


[1] : Wen, Haiguang, et al. "Deep predictive coding network for object recognition." International conference on machine learning. PMLR, 2018.\
[2] : Choksi, Bhavin, et al. "Predify: Augmenting deep neural networks with brain-inspired predictive coding dynamics." Advances in Neural Information Processing Systems 34 (2021): 14069-14083.\
[3] Marino, Joseph. "Predictive coding, variational autoencoders, and biological connections." Neural Computation 34.1 (2022): 1-44.\
[4] Boutin, Victor, et al. "Iterative VAE as a predictive brain model for out-of-distribution generalization." arXiv preprint arXiv:2012.00557 (2020).\
[5] Whittington, James CR, and Rafal Bogacz. "An approximation of the error backpropagation algorithm in a predictive coding network with local hebbian synaptic plasticity." Neural computation 29.5 (2017): 1229-1262.\
[6] Millidge, Beren, Alexander Tschantz, and Christopher L. Buckley. "Predictive coding approximates backprop along arbitrary computation graphs." Neural Computation 34.6 (2022): 1329-1368.\
[7] Tschantz, Alexander, et al. "Hybrid predictive coding: Inferring, fast and slow." arXiv preprint arXiv:2204.02169 (2022).\
[8] : Pang, Zhaoyang, et al. "Predictive coding feedback results in perceived illusory contours in a recurrent neural network." Neural Networks 144 (2021): 164-175.

**Questions:**

1) Could you detail a bit more the implementation of the FA and FFA algorithms. You are showing the losses in Figure 1, but it does not tell us how the FFA algorithm is trained. It would suggest to strongly clarify the paper to include a ‘pseudo code’ to explain the training procedure ?
2) The legend of the Figure 4 is not clear enough ! What are the rows 3 and 4 ? Is that the type of noise you apply ? The reconstruction ?
3) In Figure 5 : What is the difference between the low noise and the high noise setting ? Perceptually speaking, the noise seems to be so high in both cases that it is very hard to make the difference between both.
4) Have you controlled for the number of parameters of all compared networks ?

Minor Typo :
L 53 : critiqued —> criticized

**Limitations:**

The authors have correctly discussed the limitation part. I considered that small black and white dataset is a limitation ONLY if you compare yourself with machine learning models (which is what the authors are doing). On my opinion, this is not a limitation anymore if you leverage those ’simple’ datasets to model and probe the human cognition.

---

> ### Author Rebuttal · Authors · 2023-08-08
>
> We would like to thank the reviewer for their constructive comments and suggestions to improve the manuscript.
>
> > "Weaknesses ...”
>
> We acknowledge that a model of the brain ideally should encompass the structural constraints of the neural circuits (bio-plausible) and exhibit hallmarks of human cognition (brain-like behavior) as addressed in general response.
>
> > "Major ..1.”
>
> We thank the reviewer for pointing out the important literature, as we also acknowledged the importance of this class of models in the Introduction. However, since (Wen et al., 2018) used back-propagation through time (BPTT) to update the predictive coding network (PCN) parameters and any BPTT implementation suffers from weight transport as well as bio-implausibility specific to temporal credit assignment (Lillicrap & Santoro, 2019), we refrained from including PCN + BPTT as a benchmark.
> Regarding the decrease in accuracy over iterations as reported in Fig. 3A, we didn’t train FFA for more iterations (as opposed to PCN + BPTT). Here, we only wanted to show that even though FFA doesn’t utilize credit assignment through time, the accuracy holds up over iterations, as opposed to FA and BP which degrade performance.
>
> Predify mentioned by the reviewer also used BPTT to train feedback connections. Hence the learning algorithm severely suffers from bio-implausibility.
>
> The predictive coding formulated in (Whittington & Bogacz, 2017) and (Millidge et al., 2020) required the strict symmetry of feedforward and feedback connections at least at the initialization stage (so weight transport is still a problem in this work, see (Song et al., 2020) and the reviews). At its core, (Tscshantz et al., 2023) also uses the (Whittington & Bogacz, 2017) formulation to implement PC using Hebbian plasticity (page #8), which requires the symmetry of FF and FB connections at least at the initialization to ensure the convergence, limiting the bio-plausibility of the algorithm (In pybrid git repo, self.weight in "layers" is used in both forward and backward).
> We included these discussions to introduction thanks to the reviewer’s recommendation.
>
>
> > "2... “
>
>
> We tested the effectiveness of training for both cost functions as it is standard practice in all learning algorithm studies including the PCN studies referenced above. In sec. 4, we show the effectiveness of the model in exhibiting brain-like inferential abilities such as imagery.
>
> > "Most ...”
>
>
> We invite the reviewer to consider the heavy constraint asymmetrical weights put on FFA. FFA can optimize two cost functions AND avoid the weight transport problem. The biological brain might favor producing such a diversity of functions over raw performance on any one task. The experiments we included show the superiority of FFA over FA in performance and over BP in robustness.
>
> > "Fig 3 ...”
>
>
> We apologize for the lack of clarity. Motivated by the fact that training algorithms over time (e.g. BPTT) are complex and not easily implementable by bio-plausible circuitry, we think that the demonstration of stable recurrence without BPTT is an advance. We wanted to test to what extent each of these algorithms can maintain the visual information over iteration as a test for evaluating the learned synergy between feedforward and feedback connections in FFA.
>
> > "I would ...”
>
> We acknowledge that sensitivity to perceptual illusions and comparing to human reaction times would have been a great addition to the battery of inference tests in the paper. However, there is a tension between bio-plausibility of the learning algorithm and scalability to natural image sets. For instance,  (Pang et al., 2021) used BPTT and a pre-trained FF on ImageNet to test the ability of recurrence in producing illusory contours. Scalability is also relevant when comparison to human reaction time such as in (Tang et al., 2018). The fact that their model’s base was AlexNet (trained with BP) gave the model the ability to process natural images which were shown to human subjects when collecting reaction times. Overall, the tension between scalability and bio-plausibility has hindered filling the gap between modeling high-level cognitive abilities while being faithful to the biological constraints on neural circuitry. In our study we attempted to take the challenge of filling the gap, at least partially, by constraining the learning algorithm to biological constraints and evaluating the model in exhibiting high-level cognitive abilities such as imagery, hallucinations, and de-occlusion which has been previously studied almost exclusively in BP-trained networks (Kadkhodaie & Simoncelli, 2021; Tang et al., 2018). If anything, the dual challenge faced by the field of bio-plausibility and scalability highlights the need for further entertaining novel algorithms.
>
>
> > "​​Questions"
>
> We apologize for the oversight on clarifying the algorithm. We now included a pseudo-code in general response. Feedback alignment (FA) is an alternative learning algorithm to the backpropagation that utilizes fixed random feedback weights for error propagation removing the requirement for symmetric weight updates. FFA in essence uses FA to train both feedback and feedforward weights.
>
> > "The legend ...”
>
> We apologize for the lack of clarity of the legend of Fig.4. Please see the pdf in general response. As we stated in the first sentence of section 3.2, there is no noise used during the training.
> > "In Fig...”
>
> The noise in the legend refers to the noise in the inference algorithm that controls the convergence (Fig.1 & sec. 8.4). This experiment shows, qualitatively, that the higher noise prevents premature convergence.
> > "Have you ..”
>
> The number of trainable parameters in the FFA matches exactly to that of control BP and FA autoencoders. The feedforward control BP and FA algorithms have exactly half the number of trainable parameters as FFA by definition, since their feedback connections are not trainable.
>
>
> > "Typo..”
>
> Thank you for pointing this out.

---

> > ### Comment · Reviewer_3rRy · 2023-08-17
> > **Response to authors**
> >
> > I thank the authors for their response:
> > Concerning the literature review
> > * I admit that the PCN, predify, and PC from Whittington could not serve as a good baseline, but I still think this paper should be cited.
> > * I still think that the section 4 misses a modelling of biological data. As it is now, it seems that the presented method is not 1) showing good performance compared to ML algo (even if I definitely understand that it is hard to compete with those models because of the "heavy constraint asymmetrical weights put on FFA") and 2) not showing any good model of biological data. I just don't think that bio-plausibility is a good enough argument to justify lower performances, but the bio-plausibility should be an opportunity to model biological phenomenon. In addition, the bio-plausibility / salinity tradeoff is not a good argument : you have plenty of biological experiment that could be run with small datasets (pathfinder, path tracker...). I am not convinced by the authors's answer on this point.
> > * The pseudo definitely helps clarifying the model. Thank you !
> >
> > I am increasing a bit my rating because of the clarification, but I still think this article would benefits from more 'biological experiments'. (increasing from 3 to 4)

---

### Official Review · Reviewer_jn5o · 2023-07-05

**Soundness:** 3 good
**Presentation:** 3 good
**Contribution:** 3 good
**Rating:** 6
**Confidence:** 4

**Summary:**

The paper presents a new neural network architecture called Feedback Alignment (FFA) that demonstrates the effectiveness of co-optimizing classification and reconstruction tasks on MNIST and CIFAR10 datasets. FFA uses a goal-driven feedback pathway to alleviate weight transport problems encountered in traditional backpropagation (BP) methods, enhancing the bio-plausibility of the learning algorithm. The alignment mechanism in FFA endows feedback connections with emergent visual inference functions, including denoising, resolving occlusions, hallucination, and imagination. The study presents FFA as a promising proof-of-concept for the mechanisms underlying how feedback connections in the visual cortex support flexible visual functions and contributes to the broader field of visual inference underlying perceptual phenomena.

**Strengths:**

S1. This paper proposes a novel strategy to train neural networks to co-optimize for two objective functions based on the role of feedback connections in the brain, and leverages the credit assignment computational graph as feedback connections during learning and inference
S2. FFA for discrimination and reconstruction respectively induces noise robustness, and can flexibly support an array of versatile visual inferences such as resolving occlusion
S3. This paper is well written overall.


**Weaknesses:**

W1. Although the paper claims that FFA is bio-plausible, the discrimination and reconstruction in biological system are not processed through feedforward and feedback pathways.
W2. The datasets for test are way too simple and small.
W3. They reason for choosing discrimination and reconstruction as the tasks is not clear, as the visual system has multiple abilities.


**Questions:**

See the weaknesses stated above and also the following questions.
Q1. Please explain why FFA is bio-plausible more clearly.
Q2. Please broaden the datasets to validate the superior performance of FFA.
Q3. Some details should be added, such as how the noises are added, why choose discrimination and reconstruction as the training tasks.


**Limitations:**

While the idea is interesting, more explanations of the choices and experimental details should be added.

---

> ### Author Rebuttal · Authors · 2023-08-08
>
> > "Strengths:
> S1. This paper proposes a novel strategy to train neural networks to co-optimize for two objective functions based on the role of feedback connections in the brain, and leverages the credit assignment computational graph as feedback connections during learning and inference S2. FFA for discrimination and reconstruction respectively induces noise robustness, and can flexibly support an array of versatile visual inferences such as resolving occlusion S3. This paper is well written overall.”
>
>
> We thank the reviewer for their evaluation of our work and their helpful comments to improve the manuscript.
>
>
> > "Weaknesses:
> W1. Although the paper claims that FFA is bio-plausible, the discrimination and reconstruction in biological system are not processed through feedforward and feedback pathways.”
>
> Though interesting to consider, the above point is not readily clear to us. For clarification, is the reviewer’s point regarding the insignificance of contribution of anatomical feedforward or feedback to discrimination and reconstruction in biology (what other pathways remain? Perhaps lateral recurrence?) or is the point  about a specific lack of evidence for discrimination or reconstruction optimization in the brain?
> While we acknowledge more neural experiments are needed to shed light on the exact objectives of neural circuits and the significance of the hierarchical connections between layers to compute such objectives, we believe that modeling efforts such as the one presented here could help produce testable hypotheses for future neural experiments.
>
>
> > "W2. The datasets for test are way too simple and small.”
>
> We agree that MNIST and CIFAR10 are small datasets when compared to other AI research but in the field of models for the brain, especially under the constraints of no use of backpropagation of error, these datasets are typical for creating proof-of-concept NeuroAI research to make the foundation for future large-scale models (Lillicrap et al., 2016; Meulemans et al., 2020; Nøkland, 2016). Please see the limitation section in the paper and the scalability section in the general response.
>
> > "W3. They reason for choosing discrimination and reconstruction as the tasks is not clear, as the visual system has multiple abilities.”
>
> We agree that this is a valid question in general for studies modeling vision in the brain as they tend to mostly focus on discrimination and sometimes reconstruction (or prediction of the next stimulus). The brain exhibits many functions. Discovering these functions is an important area of research, but object discrimination is at least one of these functions and there is broad consensus that this is performed in the ventral visual stream from neural recordings (DiCarlo et al., 2012) . So we think solving a discrimination task is a good first test of any alternative to backprop. Combined with a dual task of reconstruction, this led our network to produce many other behaviors such as stability with iteration, de-noising, and imagery.
>
> > "Questions:
> See the weaknesses stated above and also the following questions. Q1. Please explain why FFA is bio-plausible more clearly. “
>
>
> We apologize for the lack of clarity on the bio-plausibility of FFA. We now added the pseudocode for FFA to make the distinction to BP. One of the key concerns for a bio-plausible implementation is that backpropagation necessitates the transmission of error signals from downstream neurons back to upstream neurons through an identical and symmetrical replica of the synaptic weight matrix that exists in the downstream path (Figure 1, BP: $W$). To elaborate further, in the process of backpropagation, the error signals (denoted as $e_f$) are multiplied by the transpose of the forward synaptic connections weight matrix ($W^T$), whereas in FFA the error signals are transported via learned existing connections $W_b$, so no other weights need to be invoked for learning besides those used in inference
>
> > "Q2. Please broaden the datasets to validate the superior performance of FFA. “
>
> This is certainly our plan to scale up to larger datasets; however, as discussed in the general response (scalability section), there are challenges shared between our algorithm and many of the bio-plausible implementations of learning that we hope will ultimately be solved with the advancement of our theoretical understanding of learning algorithms and with new architectures that could overcome the scalability issue.
>
> > "Q3. Some details should be added, such as how the noises are added, why choose discrimination and reconstruction as the training tasks.”
>
> There was no noise added during the training as indicated in the first sentence of section 3.2, and noise was used only for evaluation of noise robustness. e added the following to the caption of Figure 2:
> C) Robustness to input Gaussian noise ($\mu =0$ and varying $\sigma^2$ between 0 and 1) as measured by test accuracy on the noisy input.
>
>
>
> > "Limitations:
> While the idea is interesting, more explanations of the choices and experimental details should be added.”
>
>
>
> We thank the reviewer how found the idea interesting, we now added more explanations in the form of pseudo code on the algorithms (section 3) and on the experimental details (caption of Figure 3 and section 8.4)

---

> ### Comment · Reviewer_jn5o · 2023-08-16
>
> I appreciate the response from the authors however I still found some confusions.
> 1. In W1, to be more clear, I would like to know why the discrimination and reconstruction are processed through feedforward and feedback pathways **respectively**. Is there any evidence from biological system?
> 2. For the robustness test under noises, is that possible to add noises as in adversarial attack tasks instead of using Gaussian?
> 3. I understand the datasets are simple since the network is not large enough, could the algorithm be applied to larger networks? Also, is there a specific task that this model could solve while others cannot to show the benefits of biological plausibility?

---

> > ### Author Response · Authors · 2023-08-16
> >
> >
> > We thank the reviewers for reading through the responses.
> >
> > >” In W1,..”
> >
> > Since goal-driven feedforward neural networks with a discrimination cost function (object classification) can predict the pattern of neural activity across the hierarchy of the ventral stream (object processing part of the visual cortex), it is assumed that the discrimination is a viable loss function for the feedforward neural networks (Schrimpf et al., 2018; Yamins et al., 2014). Also, there is evidence (Kar et al., 2019; Kar & DiCarlo, 2021) that feedforward discriminative models tend to explain the earliest (presumably feedforward based) neural responses in the IT cortex.
> >
> > Regarding the feedback pathway, there is extremely scarce data from feedback neurons and to the best of our knowledge there is no direct neural predictivity mapping from DNNs to feedback pathways. From mental phenomena of imagery, there is some capacity of the system to generate visual percepts and likely this involves some form of feedback as this activity may have to propagate to parts of the early visual system (primary visual cortex) that are often required for visual awareness (Pearson, 2019). However, we can only speculate of such a division of labor (feedforward discriminative vs feedback reconstructive), and indeed these could be both integrated into the same neural population. Having two separate pathways may provide more flexibility to solve the two tasks and allows for the needed credit assignment pathways to train each other. This work suggests that feedforward and feedback neurons will be important to study to better understand their functional differences and whether that is consistent with two different optimizations.
> > Thus, although there are reasons to believe maybe discrimination is a good cost function for the forward pathway, there is not a strong experimental evidence for reconstruction as the cost function of the feedback pathway. Our results, however, show that “if” we assume reconstruction function for feedback (and alignment training with feedforward), the network exhibits brain-like visual inference properties.
> >
> > “> For the robustness...”
> >
> > In our robustness analysis, we investigated adversarial robustness as well (Supplementary material Figure 8, right panel) and we showed FFA is more robust compared to BP.
> >
> > >”I understand “
> >
> > FFA algorithm can be applied to arbitrarily large networks, however, the problem of scalability that was stated in the general response above is that we have not been able to get a performance comparable to backprop-trained networks.
> >
> > Regarding the benefits of bio-plausibility, as stated in more details in the theory section above, avoiding weight transport problem by alignment created a structural regularization effect in the network that enabled the emergence of properties of a denoiser autoencoder in supporting various visual inference tasks (resolving occlusions, imagination, …).
> >
> > Robustness induced in the FFA network may be a sign of other beneficial properties for bio-plausibility like representational straightening of natural movies (Hénaff et al., 2019, 2021; Harrington et al., 2022; Toosi & Issa, 2022) and predicting the pattern of neural activity in primate visual cortex (Dapello et al., 2020; Kong et al., 2022) both have been shown in robust neural networks.
> >
> > **References**
> >
> > Dapello, J., .., & DiCarlo, J. J. (2020). Simulating a Primary Visual Cortex at the Front of CNNs Improves Robustness to Image Perturbations. NeurIPS
> >
> > Harrington, A., ... & Freeman, W. T. (2022). Exploring perceptual straightness in learned visual representations. ICLR
> >
> > Hénaff, O. J., ... & Goris, R. L. T. (2021). Primary visual cortex straightens natural video trajectories. Nature Communications
> >
> > Hénaff, O. J., ..., & Simoncelli, E. P. (2019). Perceptual straightening of natural videos. Nature Neuroscience
> >
> >
> > Kar, K., & DiCarlo, J. J. (2021). Fast Recurrent Processing via Ventrolateral Prefrontal Cortex Is Needed by the Primate Ventral Stream for Robust Core Visual Object Recognition. Neuron
> >
> >
> > Kar, K., ... DiCarlo, J. J. (2019). Evidence that recurrent circuits are critical to the ventral stream’s execution of core object recognition behavior. Nature Neuroscience,
> >
> > Kong, N. C. L., ... & Norcia, A. M. (2022). Increasing neural network robustness improves match to macaque V1 eigenspectrum, spatial frequency preference and predictivity. PLOS Computational Biology
> >
> > Pearson, J. (2019). The human imagination: . Nature Reviews Neuroscience,
> >
> > Schrimpf, M., ...  & DiCarlo, J. J. (2018). Brain-Score: Which Artificial Neural Network for Object Recognition is most Brain-Like? BioRxiv,
> >
> >
> > Toosi, T., & Issa, E. (2022). Brain-like representational straightening of natural movies in robust feedforward neural networks. The Eleventh ICLR
> >
> > Yamins, D. L. K., Hong, H., Cadieu, C. F., Solomon, E. A., Seibert, D., & DiCarlo, J. J. (2014). Performance-optimized hierarchical models predict neural responses in higher visual cortex. PNAS

---

### Official Review · Reviewer_gcpd · 2023-07-08

**Soundness:** 3 good
**Presentation:** 3 good
**Contribution:** 2 fair
**Rating:** 6
**Confidence:** 4

**Summary:**

This paper explores the intriguing subject of integrating feedback connections into neural networks during both learning and inference stages, drawing inspiration from the brain. This study has the potential to significantly advance artificial intelligence (AI). Feedback connections are a topic of interest in both theoretical and experimental neuroscience, and establishing a link between neuroscience and AI research could mark a major advancement in both fields. Therefore, the motivation behind this study is commendable.

However, despite its compelling premise, the paper has issues with the presentation of the algorithm, its novelty, and the explanation for its superior performance over standard auto-encoders. Specifically, Figure 1 and the corresponding explanation in the main body of the text are brief and do not provide clear insight into how their system operates. Figure 1C suggests that the feedforward weights are trained using Feedback Alignment (FA) and not Backpropagation (BP) at any stage or layer of the network, yet this point is not explicitly clarified.

The authors also neglected to specify whether noise was introduced into the hidden layers to replicate the inherent noise in neurons, despite stating that no noise was added to augment the data during training. Additionally, during the inference stage, the reconstruction is updated and passed as an input in the forward pass, going through multiple iterations. However, the paper does not provide guidelines on how and when this iterative process should cease to deliver a final inference result.

In Figure 3A, there is a noticeable degradation in performance and reconstruction as a function of iterations. This degradation is less pronounced than with the FA-autoencoder or BP-autoencoder, but its presence is still problematic.

A significant point of confusion is the authors' claim that training their architecture with FA versus BP in both forward and feedback processes would lead to better generalization. This claim is not sufficiently explained in the paper. The scalability of the proposed method is another major concern. Like other FA training methods, the presented approach appears incapable of scaling to more complex problems, such as ImageNET, which is a significant limitation. Considering that there is no explanation provided for why their algorithm is more robust to noise in simpler problems like MNIST, the lack of scalability is a particular concern.

Furthermore, the claim of adversarial robustness is not adequately substantiated, as it should include a more comprehensive set of white box attacks. Since 2015, numerous studies have shown that networks improving robustness to the Fast Gradient Sign Method (FGSM) are not necessarily robust against stronger attacks, such as boundary-based optimization methods.

In summary, while the paper delves into an intriguing research area and is well-motivated, it lacks clarity regarding the proposed algorithm and its novelty compared to standard autoencoders. Additional concerns include the paper's omission of details about potential noise introduced in hidden layers, and the lack of a clear process for terminating the iterative inference stage. Additionally, the authors' assertion of better generalization through FA versus BP is not clearly explained. Issues of scalability and incomplete testing for adversarial robustness also cast doubt on the practicality and comprehensive effectiveness of the proposed method. If these issues are clearly addressed in a revision, this paper could contribute significantly to AI research."



**Strengths:**

Important problem
Interesting idea to train both forward and feedback weights with FA

**Weaknesses:**

Need more clarity
More explanation why training with FA works better for generalization than BP
Lack scalability to more complex problems
Needs stronger white box attacks to substantiate claims about robustness to adversarial attacks

**Questions:**

Explain the algorithm much better and compare to previous work like auto-encoders. Provide theoretical or intuitive explanation why training this particular architecture with FA is better than BP. Was not clear if noise is introduced in the hidden layers.

**Limitations:**

Yes

---

> ### Author Rebuttal · Authors · 2023-08-08
>
> > "Summary:
> This paper explores the intriguing subject of integrating feedback connections into neural networks....
> .”
>
>
> We would like to thank the reviewer for recognizing the significance of the aim of this work.
> We agree that the algorithm was not clarified in detail and we apologize for this oversight. We now include pseudo code (section 3) to supplement Figure 1’s visual presentation of the algorithm.
>
> > "The authors also neglected to specify whether noise was introduced ...”
>
>
> We apologize for lack of clarity on noise and the inference algorithm. We should emphasize that there was no added noise during the training. We did not model the inherent noisiness of the biological neurons. Noisiness of neurons being present in both training and evaluation presumably would require a probabilistic read-out such as what is offered in random smoothing models (Cohen et al., 2019). Thus to avoid complexity and ease of comparison to the control algorithms we avoid modeling the internet noisiness of biological neurons.
> Regarding the inference algorithm, the reviewer is absolutely correct. Although we had included the inference algorithm pseudocode to the supplementary and a schematic of inference in Figure 1, we failed to provide details on the criteria to stop the algorithm explicitly. Essentially, the algorithm stops, when the difference between the reconstructed image $\hat{x_{t-1}}$ and the input image $x_{t-1}$ falls below an imposed upper bound (sec 8.4: $ \sigma_{t-1} \leq \sigma_L $). We added details on the inference algorithm to the main paper.
>
> > "In Figure 3A, there is a noticeable degradation in performance and reconstruction as a function of iterations. This degradation is less pronounced than with the FA-autoencoder or BP-autoencoder, but its presence is still problematic.”
>
>
> The inclusion of iterations greater than 2 were for the purpose of full transparency. Since 1) in FFA the training for the reconstruction when there is no iteration (iteration 0 in the plot) and 2) FFA does not use any input noise in the training, the main purpose of the experiment was to examine the functional benefits of FFA in terms of an implicit regularization effect (denoising). The reviewer is correct that the degradation of performance over iterations happens even for FFA, but please consider the fact that this analysis is simply highlighting an emergent property of having this type of feedback to even allow for iteration – no optimization was done to handle the iterations and thus any maintenance of the performance over the iterations. It’s likely implausible to only perform end-to-end recurrence in this fashion. Future work could examine more plausible recurrent connections between each layer of the network; we simply ran the most basic one first.
>
>
> > "A significant point of confusion is the authors' claim that training their architecture with FA versus BP in both forward and feedback processes would lead to better generalization...”
>
>
> We refer to the  generalization of FFA to noise-degraded images which was shown in Figure 3. We agree that the scalability remains a limitation in this work as we addressed in the limitation section, but please refer to the general response above in the theory section for an explanation of why robustness to noise can be expected in FFA training.
>
> > " Furthermore, the claim of adversarial robustness is not adequately substantiated, as it should include a more comprehensive set of white box attacks...”
>
>
> The reviewer is absolutely correct that robustness to FGSM does not prove robustness to stronger attacks. Essentially, there are a lot of attacks and counter-attacks that have been proposed over the years in the literature to examine adversarial robustness. Here, we used FGSM as a simple yet fundamental measure of robustness that was used in previous studies of bio-plausible learning algorithms (Akrout, 2019).
>
> > "In summary, while the paper delves into an intriguing research area and is well-motivated,..."”
>
>
> We appreciate the summary of the constructive comments from the reviewer. To clarify regarding the proposed algorithm, we added a pseudocode to the paper. Regarding the novelty compared to the standard autoencoders, we should clarify that standard autoencoders use backpropagation for training which as we discussed in the paper cannot be implemented using biological neurons because of the weight transport problem. Standard denoiser autoencoders are trained using added noise to the input while the loss function includes the original noiseless image to induce robustness to noise. However, training using FFA 1) does not require weight transport hence a bio-plausible implementation and 2) does not require input noise training to induce robustness.
> We believe the reviewer is referring to noise robustness when asking about model generalization. However, in our general rebuttal above, we acknowledge that we are not entirely certain about the existence of an encompassing theory and welcome any follow-up comments if our explanation is not suitably directed to the intended generalization phenomenon.
> As we acknowledged in the limitation section of the paper, scalability is a limitation at the current stage of the work; however, as in other original, proof-of-concept work (as indicated directly in the abstract), we agree that future work relying on novel architectures and more relevant datasets and increasingly more compatible loss functions can improve the scalability to more interesting datasets. We appreciate that the reviewer could conceptualize the significance of this type of interdisciplinary research (neuroscience and AI) to the broader community of AI research.
>
> >" Weaknesses:
> Need more clarity ..."
>
> Please see the theory section in general response.
>
> >"Questions:
> Explain the algorithm ..."
>
> We now include pseudocode and intuitive theoretical accounts behind FFA training (please see general response). As stated above, no noise was introduced in the hidden layers.

---

### Official Review · Reviewer_6SSt · 2023-07-17

**Soundness:** 3 good
**Presentation:** 1 poor
**Contribution:** 4 excellent
**Rating:** 7
**Confidence:** 5

**Summary:**

The paper proposes a bio-inspired computer vision algorithm for classification that includes feedback and aims to use alignment rather than backprop for training the models of feedforward and feedback.

**Strengths:**

The paper provides a novel idea for including feedback (or recurrence) in visual classification that would be interesting for neuroscientists to investigate further.
The emerging noise robustness is a big strength for this model and confirms a lot of neuro theories about feedback.

**Weaknesses:**

The explanation of the methods is severely inadequate. The training phase is not well explained. I usually prefer to have a visual explanation over equations but in this case I believe that the learning phase, the equations for the learning process should be included as well especially given that there are variables and paramters that are included without explanation of how they contribute to the process in figure 1. The supplements also shows two separate networks for feedforward and feedback and the alignment process should be included.

The literature review is also not proper and misses many important publications in the field, for example:
- Spoerer et al 2017: Recurrent Convolutional Neural Networks: A Better Model of Biological Object Recognition
- Abdelhack and Kamitani 2018 Sharpening of hierarchical visual feature representations of blurred images
- Huang et al. 2019 Brain-inspired Robust Vision using Convolutional Neural Networks with Feedback

The first analysis shows performance on both CIFAR and MNIST while all the subsequent analyses are only including MNIST. This makes me wonder if the performance on CIFAR is not as impressive as it was not included even in supplementary materials. I think the reader would appreciate full transparency on this.

It was unclear why training on Imagenet would not be feasible. Is it a computing problem or other?

Minor: p 5 line 156 typo: discrimination -> reconstruction

**Questions:**

Are the codes going to be released?

---

> ### Author Rebuttal · Authors · 2023-08-08
>
> > " Strengths:
> The paper provides a novel idea for including feedback (or recurrence) in visual classification that would be interesting for neuroscientists to investigate further. The emerging noise robustness is a big strength for this model and confirms a lot of neuro theories about feedback.”
>
> We are extremely grateful for recognizing the novelty of the idea and for providing constructive feedback to improve the work.
>
> > " Weaknesses:
> The explanation of the methods is severely inadequate. The training phase is not well explained. I usually prefer to have a visual explanation over equations but in this case I believe that the learning phase, the equations for the learning process should be included as well especially given that there are variables and paramters that are included without explanation of how they contribute to the process in figure 1. The supplements also shows two separate networks for feedforward and feedback and the alignment process should be included.”
>
>
> We apologize for the lack of clarity in the training phase. We now included a pseudo code to the paper (section 3) to explain the alignment process.
>
> > " The literature review is also not proper and misses many important publications in the field, for example:
>
> * Spoerer et al 2017: Recurrent Convolutional Neural Networks: A Better Model of Biological Object Recognition
>
> * Abdelhack and Kamitani 2018 Sharpening of hierarchical visual feature representations of blurred images
>
> * Huang et al. 2019 Brain-inspired Robust Vision using Convolutional Neural Networks with Feedback”
>
>
> Thank you for pointing out the above list. We now added citations to the above mentioned works in literature review. However, we highlight the fact that both (Huang et al., 2019; Spoerer et al., 2017) used classical error backpropagation which as we discussed falls out of the bio-plausible implementation of learning. While (Abdelhack & Kamitani, 2018)’s finding regarding the deblurred representation in the brain is interesting, however, they don’t propose a model based on their analysis. Thus, the relevance of this paper is not immediately clear to us.
>
> > "The first analysis shows performance on both CIFAR and MNIST while all the subsequent analyses are only including MNIST. This makes me wonder if the performance on CIFAR is not as impressive as it was not included even in supplementary materials. I think the reader would appreciate full transparency on this.”
>
>
>
> As we stated in the paper (section 3.1), we only verified generative properties on the MNIST dataset  (as do virtually all similar studies on generative properties of a bio-plausible implementation) due to lack of backgrounds and less visual ambiguity in the MNIST images compared to CIFAR which makes the results of the generative processes more accessible to visual inspection . We included CIFAR results on the performance and robustness in our attempt to scale up FFA to natural images.
>
> > "It was unclear why training on Imagenet would not be feasible. Is it a computing problem or other?”
>
>
> At core, FFA is essentially two feedback alignment algorithms and although there are some theoretical attempts to explain why FA does not scale up to imagenet, there is still no single convincing theory to explain why (please refer to the general response-theory section where we explained the challenges of learning symmetric weights in the feedback). We suspect that with newer architectures maybe a variant of FA such as FFA has a chance and in future work we aim to explore this idea by implementing transformer-based architectures. A good analogy here could be the autoencoders, as for a long time it was assumed that autoencoders are unable to be scaled up to large image sets like imagenet for representational learning purposes, but masked autoencoders proved otherwise.
>
> > "Minor: p 5 line 156 typo: discrimination -> reconstruction”
>
>
>
>
> Thank you for pointing us to this typo. We now fixed this.
>
> > "Questions:
> Are the codes going to be released?”
>
>
> Absolutely, as we stated in the checklist.

---

> > ### Comment · Reviewer_6SSt · 2023-08-14
> > **Response to rebuttal**
> >
> > > We apologize for the lack of clarity in the training phase. We now included a pseudo code to the paper (section 3) to explain the alignment process.
> >
> > Thank you for the pseudocode. It makes the process much clearer now. However, it puts into question the biological plausibility since it makes the network more akin to an autoencoder since the outputs of the feedforward and the feedback passes separate. But the original paper shows in Fig1 at the inference block that the update happens on the same block. Can you please clarify?
> >
> > > Thank you for pointing out the above list. We now added citations to the above mentioned works in literature review. However, we highlight the fact that both (Huang et al., 2019; Spoerer et al., 2017) used classical error backpropagation which as we discussed falls out of the bio-plausible implementation of learning. While (Abdelhack & Kamitani, 2018)’s finding regarding the deblurred representation in the brain is interesting, however, they don’t propose a model based on their analysis. Thus, the relevance of this paper is not immediately clear to us.
> >
> > I'm surprised at this statement honestly and it reflects a lack of understanding of the context of the problem of bioplausibility of vision. For the first two references, they do implement feedback using a different algorithm yes, but they do implement feedback. You are supposed to present your work and contrast it with previous models and mention how yours is different. For the last one, since you invoke bio-plausibility as a novelty in your paper, you are supposed to cite problems with the current existing models of vision and where they fail and then contrast that with what you have. Ideally, I would have asked you to check if your model does a better job at decoding the neural representations (and that would make your paper much better and open it to more readership) but that might not be in the scope of your current paper.
> > However, if your argument is the the alignment is the only difference here and not the addition of feedback, you will have to compare your model's performance to those models with feedback such as Huang's. Currently my conclusion from your paper is that the addition of feedback is the main contribution regardless of alignment which is nonetheless a good one.
> >
> > > As we stated in the paper (section 3.1), we only verified generative properties on the MNIST dataset (as do virtually all similar studies on generative properties of a bio-plausible implementation) due to lack of backgrounds and less visual ambiguity in the MNIST images compared to CIFAR which makes the results of the generative processes more accessible to visual inspection . We included CIFAR results on the performance and robustness in our attempt to scale up FFA to natural images.
> >
> > I do not think this is a plausible justification to not even adding it to the supplementary materials. Transparency in your result presentation is always the best approach.

---

> > > ### Author Response · Authors · 2023-08-14
> > >
> > > > “...Can you please clarify?”
> > >
> > > Thank you for your response. The feedforward and feedback pathway are indeed separate. We now see that the FFA diagram in the “Learning” panel needs to show that there are two separate populations of neurons and we plan to make this change in the revision. Thanks again for pushing on this important point regarding similarity to an autoencoder.
> > > As regards the inference algorithm depicted in Figure 1, this occurs through a population of local neurons that receives input from feedback neurons and outputs (after update) are fed into the feedforward neurons of the same layer.
> > >
> > > >” I'm surprised ...”
> > >
> > > We should have clarified the scope of the feedback models we consider is “feedback for learning”, aka bioplausible backprop. The results on inference are more byproducts of this approach, but we now see more clearly the reviewer’s point that we should contextualize using both the learning and inference literature as the reader is not as a priori cued into the thrust of our work. In a revision, we can make this clearer in general by mentioning models with feedback for inference in a separate paragraph and going over the advances feedback provides and how we wanted to take a first attempt at demonstrating those phenomena in our bio-plausibly trained network. We appreciate the reviewer’s very informed perspective based on a broad understanding of the latest literature on biological vision models using feedback. And we too would like to compare to neural data in the future, but our models as well as others do not quite scale up enough to compare to real brains. One main challenge for predicting neural data in feedback models is the scarcity of neural data from feedback neurons, however, these modeling efforts could inform neural experiments in identifying the properties of feedback vs feedforward neurons.
> > > Our original manuscript focussed on models for biologically plausible backprop; hence why we chose the control algorithms such as BP and FA as they use feedback for credit assignment and they are the control algorithms in previous similar studies (Gilmer et al., 2017; Liao et al., 2016; Lillicrap et al., 2020; Moskovitz et al., 2018; Nøkland, 2016). Guided by the reviewers’ comment, we realize that maybe we should have relaxed this assumption and added as many as models of feedback for inference including (Huang et al., 2020) and we would have discussed the similarities of differences in the context of architecture, learning algorithms and different loss functions. However, (Huang et al., 2020). was a short workshop paper with limited methodological details and that has no available code as far as we can find, so there are few options on feedback based inference networks, and the autoencoder seemed like a tried and true comparison point in the interim.
> > >
> > > Thank you for the clarification regarding predicting the neural representations in (Abdelhack & Kamitani, 2018), we didn’t realize that the fMRI data from that paper is available and we agree that it would have been interesting and insightful if we were able to make image-computable predictions for their data using our model. At minimum, we should discuss this paper’s finding as a motivating one for studying inference under ambiguity if we are to truly match the strengths of biological vision with new machine vision models.
> > >
> > > >”However, ...”
> > >
> > > Indeed our results show that the learning algorithm (alignment) created the noise robustness effect (the link was discussed in the theory section of general response) and hence the flexible visual inference capability. However, we acknowledge that despite the implausibility of the core learning algorithm (BP) in (Huang et al., 2020), exploring the inference properties would have been interesting since it also exhibited noise robustness. Comparing these two could also be insightful in terms of theoretical explanation of the benefits of alignment.
> > >
> > > >”I do not think ...”
> > > We agree with the reviewer that transparency is the best approach and that is why the code for CIFAR implementation was already in our repo for reproducibility. We will add the CIFAR10 results of inference to the supplementary of the camera ready version should the paper gets accepted.
> > >
> > > We agree on the reviewer’s point about transparency; we now recognize that it is important to include the full set of CIFAR results. Thanks to the reviewer for pointing this out in their feedback. We are happy to run the inferences and include them in the revised version.
> > >
> > >
> > > Lillicrap, T. P., Santoro, A., Marris, L., Akerman, C. J., & Hinton, G. (2020). Backpropagation and the brain. Nature Reviews Neuroscience
> > >
> > > Liao, Q., Leibo, J. Z., & Poggio, T. (2016). How important is weight symmetry in backpropagation?
> > >
> > > Moskovitz, T. H., Litwin-Kumar, A., & Abbott, L. F. (2018). Feedback alignment in deep convolutional networks. ArXiv
> > >
> > > Nøkland, A. (2016). Direct Feedback Alignment Provides Learning in Deep Neural Networks. ArXiv

---

> > > > ### Comment · Reviewer_6SSt · 2023-08-15
> > > >
> > > > Just a quick comment on the paper by Huang 2019 (not 2020). The code is released here: https://github.com/yjhuangcd/CNNF

---

> > > > > ### Author Response · Authors · 2023-08-15
> > > > >
> > > > > Thank you for directing us to their Git repository; it significantly simplifies the process of reproducing the results and conducting comparisons to Huang et al 2019.

---

### Official Review · Reviewer_KkB4 · 2023-07-18

**Soundness:** 3 good
**Presentation:** 2 fair
**Contribution:** 3 good
**Rating:** 6
**Confidence:** 4

**Summary:**

The authors propose a mechanism for biologically plausible learning in visual circuits that can endow the system with additional epiphenomenal capabilities that resemble phenomena humans are known to exhibit. The mechanism resembles feedback alignment, where the weight transport problem is resolved by substituting different feedback weights in the backward pass instead of the transpose of the forward weights. In contrast to previous approaches that dynamically train the backward weights to achieve symmetry through local learning, the authors suggest concurrently training the feedback weights to reconstruct the system's visual inputs based exclusively on activations in the final latent layer. They then show that these feedback weights can be used for a wide variety of additional tasks (similar to Kadkhodaie & Simoncelli 2021), including image generation, conditional image generation based on a weakly clamped latent layer, and denoising.

**Strengths:**

The concept for the paper is a very strong one: it is a major failing of the 'feedforward paradigm' of visual cortex modeling that it cannot account for capabilities and phenomena the visual system is known to exhibit, including mental imagery, hallucination, and to some extent de-occlusion (e.g. filling in the visual blind spot caused by the optic nerve). That a network can be shown to accomplish this through exclusively through a reasonably local learning algorithm (a feedback alignment variant) is very impressive. Furthermore, the notion that the feedback pathway could have additional functional utility beyond exclusively providing credit assignment signals for learning is a much-needed perspective for the biologically plausible plasticity field, especially the subfield that is primarily focused on variants of the feedback alignment algorithm.

**Weaknesses:**

On the theory of learning side of things, the paper falls short in several ways.

First and foremost, the authors do not actually explain their learning algorithm in the detail that would be required to reproduce it. The authors should provide detailed math explaining the actual steps required to implement the proposed rule at each layer of their system, as well as potentially pseudocode. This is required for me to increase my confidence and maintain my score--what I imagine the authors to be doing based on my own understanding of the feedback alignment algorithm is a perfectly viable approach, but it's not possible to confirm this based on the text itself. For reproducibility purposes, the authors should also commit to providing a github link to their code implementation of the model.

Second, (as with the original FA paper) it is not obvious why the authors' approach works at all. It is possible to draw some loose similarities between the proposed algorithm and target propagation, but the authors provide no mathematics that explain why the proposed algorithm should actually produce learning. Given that the empirical performance results begin to drop off steeply for CIFAR10 (and presumably fail entirely for imagenet), the convergence properties of this algorithm deserve serious attention.

Furthermore, the authors do not perform any form of direct comparison to alternative biologically plausible algorithms that would be expected to show similar results. In particular, both the predictive coding (PC) approach and the Wake-Sleep algorithm should show similar effects, and energy-based models like the Boltzmann machine, PC, and even Hopfield networks have a long history of demonstrating pattern completion effects and image generation on simple datasets like binarized MNIST. The paper would benefit immensely from a discussion of the experimentally testable predictions that are specific to this model and not to other approaches, as well as a discussion of the factors that make this method conform to known experimental data more than these methods.

Lastly, the inpainting effects produced in this paper are qualitatively worse than those observed in Kadkhodaie & Simoncelli 2021. The paper would benefit from an empirical comparison to the model actually used in this paper, or at least a discussion of the similarities & differences, as well as why we should expect worse results here.

**Questions:**

Do multilayer perceptron networks trained on these tasks show qualitatively similar inpainting and generation? The authors say that the MLP results are present, but I've been unable to find them in the text or supplemental. Networks that do not have within-layer weight sharing are of more interest from the perspective of biological plausibility, and while I believe that an MLP should produce similar results, it would be good to have visual confirmation.

Could you discuss the biological plausibility of BatchNorm, which appears to be used in all tested network architectures? Have similar results been obtained in the absence of BatchNorm?

**Limitations:**

Yes, the authors do adequately discuss the limitations of their work.

---

> ### Author Rebuttal · Authors · 2023-08-08
>
> > "
> Strengths:
> The concept for the paper is a very strong one: it is a major failing of the 'feedforward paradigm' of visual cortex modeling that it cannot account for capabilities and phenomena the visual system is known to exhibit ..."
>
> We appreciate that the reviewer recognizes the significance of the concept of this work and we thank the reviewers for their inputs on our work.
> > "
> Weaknesses:
> First and foremost, the authors do not actually explain their learning algorithm in the detail that would be required to reproduce it..."
>
> We apologize for the lack of clarification regarding the learning algorithm. We now added the learning algorithm as pseudo codes to the paper (and in the general response above). As we stated, we will provide the github link to our code including training models by FFA and control algorithms.
> > "
> Second, (as with the original FA paper) it is not obvious why the authors' approach works at all. It is possible to draw some loose similarities between the proposed algorithm and target propagation..."
>
> Please see the theory section and scalability section in the general response where we addressed this concern. We add here that TP invokes targets at each layer whereas we uniquely only set targets at the discriminator and reconstructor outputs allowing for end-to-end training of the dual loss. End-to-end training tends to perform well relative to local losses with the drawback of requiring backprop, but we benefit from end-to-end goals in a bio-plausible way by learning the backpropagation path implicitly through alignment and not having to invoke a separate, symmetric pathway.
>
> > "Furthermore, the authors do not perform any form of direct comparison ..."
>
> Regarding bio-plausibility of algorithms, the premise of our work was to discover a bio-plausible learning algorithm for mainly feedforward deep neural networks which share many neuronal and behavioral similarities to biological vision (Schrimpf et al., 2018, 2020). The generative properties of the implementation were not a direct aim of our set-up but illustrate downstream consequences of this type of feedback, and we tried to stay away from any claims regarding the superiority of FFA in terms of pattern completion or generation. Rather, the fact that feedforward and feedback pathways used for training each other can give rise to generative properties serves as a proof-of-concept illustrating the utility of bio-plausible implementation. Though theoretically intriguing, Hopfield networks, Boltzmann machine and Predictive coding approaches in general, lag behind the DNN class of models in classification performance ((Belyaev & Velichko, 2020) reports a Hopfield network with 56.2% on MNIST) and are have poor predictivity of neural responses and human behavior (Schrimpf et al., 2018) (brain-score.org).
>
> > "Lastly, the inpainting effects produced in this paper are qualitatively worse than those observed in Kadkhodaie & Simoncelli 2021. The paper would benefit from an empirical comparison to the model actually used in this paper, or at least a discussion of the similarities & differences, as well as why we should expect worse results here."
>
>
> Kadkhodaie and Simoncelli 2021 trained autoencoders using backpropagation to be robust to input noise by including noisy images in the training set. As we pointed out in the paper (section 3.2), networks trained using FFA were never shown noisy images during training, and we only used noise for evaluation of robustness and inference algorithm. The point we make in the paper is that FFA induces the properties of a denoising autoencoder (presumably by a form of structural regularization, please see the theory section in general response) without receiving any noise training, even if weaker than the denoising autoencoder trained with BP.
>
> > " Questions:
> Do multilayer perceptron networks trained on these tasks show qualitatively similar inpainting and generation? The authors say that the MLP results are present, but I've been unable to find them in the text or supplemental. Networks that do not have within-layer weight sharing are of more interest from the perspective of biological plausibility, and while I believe that an MLP should produce similar results, it would be good to have visual confirmation."
>
> We acknowledge the ambiguity present in the text. We tested both fully connected (labeled as customized architecture) and resnet-like convolutional architectures for the co-optimization verification (section 3). However, the results in the visual inference were on the convolutional architecture. We agree that weight-sharing in a convolution module has been raised as a concern for bio-plausibility although MLP models which allow connection of any neuron in a layer to any other neuron is also problematic from the biological perspective (neurons tend to send projections to nearby neurons and connectivity strength falls quickly over distance). Also, the theoretical foundation of sampling implicit prior does not assume any architectural constraints (Kadkhodaie & Simoncelli, 2021) confirming the reviewer’s intuition regarding MLP should produce similar results. However, we plan to add this to the supplementary should the paper get accepted.
>
> > "Could you discuss the biological plausibility of BatchNorm, which appears to be used in all tested network architectures? ...?"
>
>
> We agree that BatchNorm in the current form can hardly be considered a bio-Plausible implementation and in future work we plan to replace that with closely related but bio-inspired modules such as (Shaw et al., 2020). BatchNorm has been used in other research related to bio-plausibility of learning algorithms (Kunin et al., 2020; Liao et al., 2016), presumably with the same logic that it is rather feasible to replace this module with a more bio-plausible implementation as the community discovers more about BatchNorm and the biological mechanisms underlying the regularization in the brain.

---

> > ### Comment · Reviewer_KkB4 · 2023-08-10
> > **Response to authors**
> >
> > Thank you for your detailed feedback. The authors' commitment to include algorithm pseudocode and as well as the actual code used to generate the simulations has caused me to maintain my score. The remainder of the authors' comments have confirmed my initial belief in my score (6) and motivate me to increase my confidence (4).
> >
> > One small note: the authors' argument that Boltzmann machine and predictive coding-style approaches lag behind algorithms that explicitly approximate backpropagation is unconvincing to me, especially since their own results have been essentially isolated to MNIST. A discussion of similarities and differences that would be observed or tested at a cellular level and deal with the actual dynamics of synaptic plasticity would be more convincing than an appeal to the types of network-wide steady-state effects typically reported by BrainScore, which are sensitive to the neuron model used and many features of the objective function & learning scheme (choice of optimizer) that do not necessarily have to do with the actual learning mechanism used by the system.

---

> > > ### Author Response · Authors · 2023-08-10
> > >
> > > We would like to thank the reviewer for reading our responses and increasing their confidence. The confidence score that the system show is still 3, so we assume it may take some time to get updated.
> > >
> > > Thank you for the clarification regarding alternative algorithms, point is well-taken.

---

### Author Rebuttal · Authors · 2023-08-08

We thank the reviewers for their helpful comments to improve the work. Below, we address the general theme of concerns which were shared between reviewers.

**FFA algorithm in pseudo code**


Weight updates in a three-layer linear network:
parameters: $W_{f_1}$, $W_{f_2}$, $W_{b_1}$, $W_{b_2}$ \
for $epoch<n_{epoch}$: \
&nbsp;&nbsp;&nbsp;&nbsp;$Y = W_{f_2} \cdot h_f; h_f= W_{f_1} \cdot X$ \
&nbsp;&nbsp;&nbsp;&nbsp;$e_f = T - Y$ \
&nbsp;&nbsp;&nbsp;&nbsp;$Loss_f = \frac{1}{2} e_f^T \cdot e_f$ \
&nbsp;&nbsp;&nbsp;&nbsp;$\Delta W_{f_2} = -e_f^T \cdot h_f^T,$ $\Delta W_{f_1} = -W_{b_2} \cdot e_f \cdot X^T$
    (forward updates) \
&nbsp;&nbsp;&nbsp;&nbsp;$\hat{X} = W_{b_1} \cdot h_b; h_b = W_{b_2} \cdot Y$\
&nbsp;&nbsp;&nbsp;&nbsp;$e_b = X - \hat{X}$ \
&nbsp;&nbsp;&nbsp;&nbsp;$ Loss_b = \frac{1}{2} e_b^T \cdot e_b$ \
&nbsp;&nbsp;&nbsp;&nbsp;$\Delta W_{b_1} = -e_b^T \cdot h_b^T,  \Delta W_{b_2} = -W_{f_1} \cdot e_b \cdot Y^T$
    (backward updates)

**The Theoretical motivation for FFA**

FFA at its core uses Feedback alignment (FA). As pointed out by Reviewer KkB4, the theory behind why FA works is still under active research (Cheng & Brown, 2023; Jan, n.d.; Refinetti et al., 2021; Robertson & Koyejo, 2023)
As we highlight in our abstract, this work aims to provide a proof-of-concept for how bio-plausible implementation of learning through feedback connections can support versatile visual inference. However, there were theoretical motivations behind the creation of this algorithm.


**Regularized autoencoders arrive at symmetric solutions.**
*(rationale on the link between the choice of reconstruction loss and bio-plausibility of training)*
As discussed in the Introduction, the weight transport problem of bio-plausibility of backpropagation points to the fact that backpropagation requires symmetric weights in the forward and the backward computational graph ($W_f$ and $W_f^T$ in Figure 1). If there was a way to learn to copy the weights from the feedforward to the feedback connections in between weight updates, then this problem would be alleviated. A theoretical work (Kunin et al., 2019) shows that in autoencoders which were trained under $L_2$ regularization, decoder and encoder are the transposes of each other.
We reasoned that if instead of the fixed random weights in the backward pass of FA, we let the backward pass to reconstruct what the forward pass generates, then essentially we are encouraging the learned weights to be transposes of each other and thus be a better approximation to backpropagation.
Thus, apart from the unsupervised nature of a reconstruction loss, and the structural similarity of feedback and feedforward, arriving at symmetric solutions to better estimate backpropagation was another motivation for reconstruction loss for the feedback pass.


**Theoretical Analogy of structural regularization to input noise robustness**
*(why FFA training induces noise robustness?)*


In the paper, we showed that although FFA didn’t use any noise during training, it still exhibits robustness to input noise. We later showed that this robustness to noise enables FFA to exhibit properties of a denoiser autoencoder in using the learned implicit prior (Kadkhodaie & Simoncelli, 2021) to support various generative properties. Thus, it is important to understand the link between FFA and robustness to input noise. A classical work by Bishop (Bishop, 1995) showed that training neural networks under Tikhonov regularization is mathematically equivalent to training these networks to be robust to input noise. In essence, (Bishop, 1995) provides the link between a structural regularization and input noise robustness. In FFA, feedforward and feedback connections updates are highly constrained to each other and we think that this could have served as a regularization on the weights. However, proving such a link would be a significant undertaking and we acknowledge that such a theoretical link should be proven in our future work.




**Scalability.** As addressed in Limitations, FFA has problems with the architectures where the dimension of representation reduces drastically over depth, bearing some resemblances to what was previously reported for FA. Some form of Transformers could probably do better, however, the challenges of implementing the modules with a trainable backward computational graph hinders this line of reach from benefiting from autograd packages.


**“Bio-plausible” learning algorithms and “Brain-like” visual inferences**
 Bio-plausible is an umbrella term used for structural, algorithmic, or conceptual theories about the brain. In our work, we used bio-plausible to refer to the plausibility of implementation of learning in biological neural networks, i.e. avoiding the requirement to copy the weights of feedforward to feedback after each weight update (weight transport problem).


* no weight transport is needed in our learning algorithm


* the reciprocal connections between layers of the hierarchy (Markov et al., 2013; Rockland, 2022)


Among the bio-plausible algorithms proposed in past works, only FA has both of these criteria. However, to be able to have a reference to compare to other non-bio-plausible algorithms, we picked BP, the main algorithm for training DNNs. It is worth mentioning that this is a typical choice of control algorithms in the literature for bio-plausible learning algorithms (Lillicrap et al., 2016; Bengio, 2014; Goudarzi, 2017)


By brain-like, we referred to solving inference problems that humans appear to do effortlessly such as accurate perception even in presence of occlusions, and forming mental imagery, hallucination, etc. Predictive coding came up as a bio-plausible algorithm in several reviews. As we wrote in detail in addressing Reviewer 3rRy’s concerns, most existing implementations of the predictive coding framework use BP and BP-through-time to update the weights which violates the bio-plausibility of implementation we aimed for.

---

### Decision · Program_Chairs · 2023-09-21

**Decision:**

Accept (poster)

**Comment:**

Inspired by the feedback connection in the brain, this paper proposes a biologically plausible learning method that concurrently optimizes feedforward and feedback weights with different but dependent objectives for discrimination and reconstruction tasks to induce robustness to image noise and adversarial attacks. After rebuttal and discussions, most reviewers agreed that the topic of this paper is very important for both AI and neuroscience research and the proposed idea is interesting and novel. Two reviewers giving a final rating of 4 also acknowledged their concerns addressed and improved their original ratings. The AC looked through the paper and all the comments and agreed that the paper should be accepted. The AC suggested the authors fix all the issues mentioned in the discussions, including but not limited to: (1) unclear presentation – there should be enough formulations or pseudo code to clarify the detailed steps of the method, and the figures should be polished to be more informative; (2) complementing the literature review, especially for the ones listed by reviewers; (3) adding more explanations or discussions regarding the questions raised by reviewers.